# The *Pseudomonas aeruginosa* substrate-binding protein Ttg2D functions as a general glycerophospholipid transporter across the periplasm

Daniel Yero [1,2], Mireia Díaz-Lobo[3], Lionel Costenaro [1], Oscar Conchillo-Solé [1,2], Adrià Mayo[1], Mario Ferrer-Navarro[1], Marta Vilaseca [3], Isidre Gibert [1,2✉] & Xavier Daura [1,4✉]

In *Pseudomonas aeruginosa*, Ttg2D is the soluble periplasmic phospholipid-binding component of an ABC transport system thought to be involved in maintaining the asymmetry of the outer membrane. Here we use the crystallographic structure of Ttg2D at 2.5 Å resolution to reveal that this protein can accommodate four acyl chains. Analysis of the available structures of Ttg2D orthologs shows that they conform a new substrate-binding-protein structural cluster. Native and denaturing mass spectrometry experiments confirm that Ttg2D, produced both heterologously and homologously and isolated from the periplasm, can carry two diacyl glycerophospholipids as well as one cardiolipin. Binding is notably promiscuous, allowing the transport of various molecular species. In vitro binding assays coupled to native mass spectrometry show that binding of cardiolipin is spontaneous. Gene knockout experiments in *P. aeruginosa* multidrug-resistant strains reveal that the Ttg2 system is involved in low-level intrinsic resistance against certain antibiotics that use a lipid-mediated pathway to permeate through membranes.

[1] Institut de Biotecnologia i de Biomedicina (IBB), Universitat Autònoma de Barcelona (UAB), Barcelona, Spain. [2] Departament de Genètica i de Microbiologia, UAB, Barcelona, Spain. [3] Institute for Research in Biomedicine (IRB Barcelona), The Barcelona Institute of Science and Technology, Barcelona, Spain. [4] Catalan Institution for Research and Advanced Studies (ICREA), Barcelona, Spain. ✉email: Isidre.Gibert@uab.cat; Xavier.Daura@uab.cat

**P**seudomonas aeruginosa are among the most important multidrug-resistant (MDR) human pathogens[1], showing inherent resistance to an important fraction of the available antibiotics[2]. P. aeruginosa are responsible for chronic lung infections in individuals with chronic obstructive pulmonary disease or cystic fibrosis (CF)[3] and account for over a tenth of all nosocomial infections[4]. A number of effective drugs and formulations can treat P. aeruginosa infections, even in CF patients[5]. These include frontline antibiotics such as piperazillin-tazobactam, ceftazidime, aztreonam, imipenem, meropenem, ciprofloxacin, levofloxacin, tobramycin, amikacin, and colistin[6]. Yet, resistance to most of these antimicrobials is being increasingly reported[7]. The basis for the inherently high resistance of these microorganisms is primarily their low outer-membrane (OM) permeability[8,9], complemented by the production of antibiotic-inactivating enzymes (e.g. β-lactamases), the constitutive expression of efflux pumps[10,11], and the capacity to form biofilms[1,12], among other mechanisms. The susceptibility of P. aeruginosa to antimicrobials can be additionally reduced by the acquisition of inheritable traits, including horizontal gene transfers and mutations that decrease uptake and promote efflux pump overexpression[13–15]. Although a number of genes and mechanisms of resistance to antibiotics are already known in P. aeruginosa, the complex mechanisms controlling the basal, low-level resistance to these compounds are still poorly understood[16,17].

The OM of P. aeruginosa is known to be central to its antibiotic-resistance phenotype. Its intrinsically low permeability is partly determined by inefficient OM porin proteins that provide innate resistance to several antimicrobial compounds, mainly of hydrophilic nature[1,8,10]. In addition, studies with mutant strains have shown that the loss of specific efflux pump mechanisms, commonly overproduced in clinical isolates, is compensated by reducing the permeability of the OM[9]. Furthermore, polymyxin resistance in P. aeruginosa is associated with significant alteration of the membrane's glycerophospholipid composition[18]. Thus, mechanisms involved in OM organization, composition, and integrity interfere with the diffusion through the membrane of both hydrophilic and hydrophobic antimicrobial compounds. The asymmetry in the lipid organization of the OM is a main factor in the low permeability to lipophilic antibiotics and detergents[19]. Glycerophospholipid trafficking across bacterial membranes contributes to that asymmetry, but the systems involved in this transport and their directionality are just beginning to be understood.

In Escherichia coli, the Mla system (MlaA-MlaBCDEF) was initially proposed to have a phospholipid import function, preventing phospholipid accumulation in the outer leaflet of the OM and thus controlling membrane-phospholipid asymmetry[20]. The core components of this ATP-binding-cassette (ABC) transport system in the inner membrane (IM) comprise the permease (MlaE), the ATPase (MlaF), and the substrate-binding protein (SBP) MlaD that are highly conserved among Gram-negative bacteria[21]. The MlaA component, an integral OM protein that forms a channel adjacent to trimeric porins, would selectively remove phospholipids from the outer OM leaflet and transfer them to the soluble periplasmic SBP MlaC[22,23]. MlaC would then transport the phospholipids across the periplasm and deliver them to MlaD for active internalization through the IM[24]. Deletion of the genes of this system is known to destabilize the OM, and bacterial strains lacking any of the Mla components are more susceptible to membrane stress agents[20,25–31]. More recently, the retrograde transport hypothesis has been questioned and a new role for this system in anterograde phospholipid transport has been suggested. Thus, the Mla system in E. coli and its homolog in Acinetobacter baumannii appear to participate

in the export of phospholipids to the OM[30,32]. In this context, MlaD would extract phospholipids from the IM and transfer them to MlaC in an ATP-independent manner.

The orthologous Mla system in P. aeruginosa is encoded by the PA4452–PA4456 operon (locus tags corresponding to PAO1) and the isolated gene PA2800 (MlaA ortholog, also known as VacJ). Proteins encoded by this gene cluster are highly similar to those encoded by operon ttg2 (toluene tolerance genes) in Pseudomonas putida[33,34]. Although it is unlikely that organic solvents themselves are substrates of this transporter, this system was initially linked to toluene tolerance in this species[35]. Accordingly, components of the P. aeruginosa ABC transporter encoded by the PA4452–PA4456 have been named Ttg2A (MlaF), Ttg2B (MlaE), Ttg2C (MlaD), Ttg2D (MlaC), and Ttg2E (MlaB)[33]. Recent studies of mutant strains with disrupted ttg2 or vacJ genes support the contribution of this ABC transport system to the intrinsic resistance of P. aeruginosa to antimicrobials[25,29,33,36]. Nevertheless, one of these studies has challenged the role of this system in intermembrane phospholipid trafficking in P. aeruginosa[33].

Here, we have primarily focused on P. aeruginosa's Ttg2D (Ttg2D$_{Pae}$) and the nature of its cargo, which shall provide additional clues on the function of the Ttg2 system as a whole. Thus, we present structural and functional evidence of the role of this protein as a general phospholipid transporter capable of carrying either diacyl or tetra-acyl glycerophospholipids. Our structural analysis further enriches the existing knowledge on the structural diversity of SBPs and supports current discussions on the directionality of phospholipid transport by the Mla system. In addition, based on mutational studies of the ttg2 operon, we have validated the contribution of the Ttg2 system to the intrinsic basal resistance of P. aeruginosa to several antibiotic classes and other damaging compounds. Although the role of other components of this ABC transport system in multidrug resistance has been already established for P. aeruginosa[25,33], this is the first study focusing on the soluble periplasmic SBP component, Ttg2D$_{Pae}$. Among the components of the Ttg2 system, this SPB could be the most promising candidate for an antimicrobial intervention based on the specific blocking of this trafficking pathway.

## Results

**Ttg2D$_{Pae}$ contains a large hydrophobic cavity that binds four acyl tails.** Sequence analysis indicates that Ttg2D$_{Pae}$ (PA4453) is the soluble periplasmic SBP component of the ABC transporter encoded by the ttg2 operon and a member of the Pfam family MlaC (PF05494). Interestingly, the available three-dimensional (3D) structures for the MlaC family from Ralstonia solanacearum (PDB entry 2QGU), P. putida (PDB entries 4FCZ and 5UWB), and E. coli (PDB entry 5UWA) were all solved in complex with a ligand in their hydrophobic pocket, except for one structure from E. coli (PDB entry 6GKI) where the protein was delipidated. Electron densities for the ligand were in all these structures compatible with phospholipid moieties, supporting their predicted role as phospholipid transporters. A sequence alignment shows that some of the residues thought to be involved in phospholipid binding in the R. solanacearum Ttg2D structure are conserved in the P. aeruginosa ortholog (Supplementary Fig. 1). Remarkably, the electron densities for P. putida Ttg2D revealed the presence of two diacyl lipids in its pocket[27].

To investigate ligand binding at the molecular level, we determined by molecular replacement the crystallographic structure of the Ttg2D$_{Pae}$ mature protein (without the signal peptide, aa 23–215) produced in E. coli at 2.53 Å resolution (PDB entry code 6HSY) (Fig. 1a). The structure was refined to a final $R_{work}$ and $R_{free}$ of 20.9 and 24.9%, respectively, and good

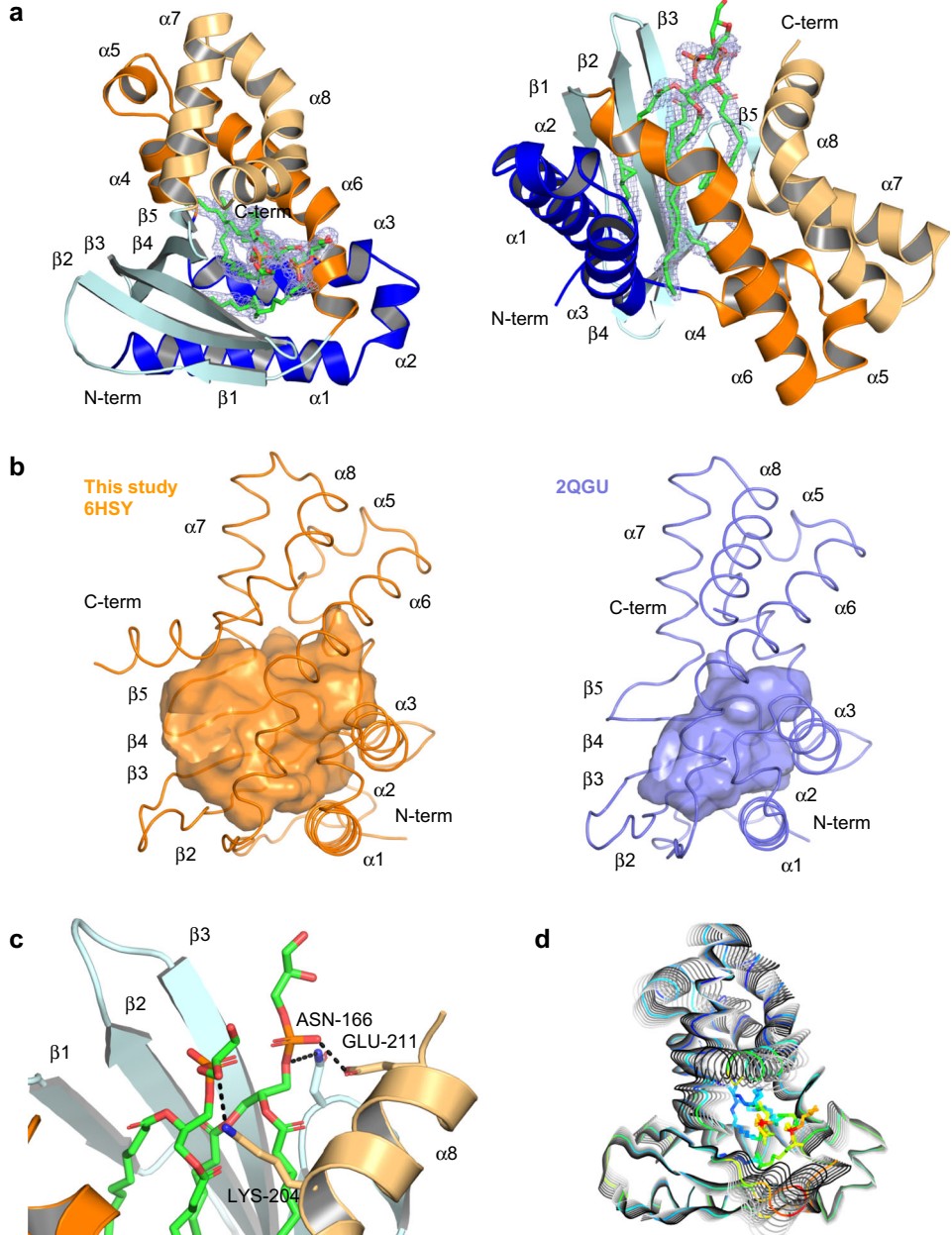

**Fig. 1 Ttg2D$_{Pae}$ binds two phospholipids simultaneously. a** Crystal structure of Ttg2D$_{Pae}$ with two PG (16:0/cy17:0) bound. The feature-enhanced electron-density map around the modeled lipids, shown as a mesh, is contoured at 1.5$\sigma$. The cartoon representation of the protein is colored according to the CATH domains: domain 1 in blue and domain 2 in orange, dark tones for segments 1 and light tones for segments 2 in each domain (see Supplementary Fig. 1). Left panel emphasizes the "decanter" shape of the structure, and right panel the lipid binding. **b** Structures of Ttg2D from *P. aeruginosa* (orange, left panel) and *R. solanacearum* (slate, right panel). The cavities of both proteins are shown as semi-transparent surfaces (side view from the right of **a**). **c** Interactions between the lipid head group and the protein. Hydrogen bonds are shown as black dotted lines. **d** Protein motions along the normal mode 7. The colors represent the *B*-factors (spectrum blue to red for lowest to highest values).

validation scores (Table 1). All residues but the last three C-terminal ones (plus the C-terminal expression tag) could be modeled. Ttg2D$_{Pae}$ adopts a mixed $\alpha + \beta$ fold with a highly twisted anti-parallel β-sheet formed by five strands and surrounded by eight α-helices. It exhibits a "decanter" shaped structure never described before for any other protein family (Fig. 1a). The structure presents a highly hydrophobic cavity between the β-sheet and the helices that spans the whole protein and has a volume of 2979 Å$^3$ and a depth of ~25 Å (Fig. 1b). After the first refinement stage (AutoBuild), without any ligand added, clear density was visible inside the cavity that could correspond to

four acyl chains (Supplementary Fig. 2). We therefore modeled inside the cavity two PG(16:0/cy17:0) (Fig. 1a), as mass spectrometry (MS) experiments suggested that this lipid was one of the most abundant among those found to bind Ttg2D$_{Pae}$ when expressed in *E. coli* (see below). Real-space correlation coefficients of 0.9 for the lipids indicate a good fit to the $2mF_o - DF_c$ electron density. The four acyl tails are deeply inserted into the hydrophobic cavity, while the polar head groups are exposed to the solvent and make only few contacts with the protein (Fig. 1a, c). This lack of specific recognition of the head group could explain why Ttg2D$_{Pae}$ is able to bind different types of

**Table 1 Data collection and model refinement statistics of the crystal structure.**

| | Ttg2D—two PG (16:0/cy17:0) |
|---|---|
| Data collection | |
| Wavelength (Å) | 0.9802 |
| Space group | $P3_221$ |
| Unit cell parameters (Å) | $a = b = 124.64$, $c = 38.06$ |
| Resolution range (Å) | 62.32–2.53 (2.64–2.53)[a] |
| No. of reflections | |
| Total | 71,596 (6246) |
| Unique | 11,489 (1349) |
| Completeness (%) | 99.6 (98.0) |
| Average multiplicity | 6.2 (4.6) |
| $<I/\sigma(I)>$ | 8.2 (2.0) |
| $R_{meas}$ (%)[b] | 13.5 (79.5) |
| $R_{pim}$ (%)[c] | 5.3 (35.2) |
| $CC_{1/2}$ (%)[d] | 99.6 (71.7) |
| $B$-factor from Wilson plot (Å$^2$) | 43.0 |
| Model refinement | |
| No. of reflections used | 11462 |
| $R_{work}/R_{free}$ (%)[e] | 20.9/24.9 |
| No. of non-H atoms (all/protein/ligands/water) | 1675/1527/127/21 |
| No. of protein residues/chain per a.u. | 190/1 |
| RMS deviations | |
| Bond lengths (Å) | 0.003 |
| Bond angles (°) | 0.47 |
| Average $B$-factors (Å$^2$, all/protein/ligands/water) | 68.0/66.9/83.2/61.0 |
| Molprobity scores | |
| Overall score, %ile | 0.78, 100th |
| Clashscore, %ile | 0.31, 100th |
| Poor rotamers (%) | 1.21 |
| Ramachandran outliers (%) | 0 |
| Ramachandran favored (%) | 97.87 |
| PDB code | 6HSY |

[a]Values in parentheses are for the highest resolution shell.
[b]$R_{meas} = \Sigma_\mathbf{h} (n_\mathbf{h}/(n_\mathbf{h} - 1))^{1/2} \Sigma_i |I_{\mathbf{h},i} - <I_\mathbf{h}>|/\Sigma_\mathbf{h}\Sigma_i I_{\mathbf{h},i}$, where $n_\mathbf{h}$ is the number of observations of reflection $\mathbf{h}(hkl)$, $I_{\mathbf{h},i}$ the $i$th measurement of its intensity and $<I_\mathbf{h}>$ the average of all $I_{\mathbf{h},i}$.
[c]$R_{p.i.m.} = \Sigma_\mathbf{h} (1/(n_\mathbf{h} - 1))^{1/2} \Sigma_i |I_{\mathbf{h},i} - <I_\mathbf{h}>|/\Sigma_\mathbf{h}\Sigma_i I_{\mathbf{h},i}$.
[d]Correlation coefficient between intensities from random half-data sets.
[e]$\Sigma_\mathbf{h} |F_o| - |F_c|| / \Sigma_\mathbf{h}|F_o|$, where $|F_o|$ and $|F_c|$ are observed and calculated structure factor amplitudes, respectively. $R_{free}$ was calculated using 5% of the reflections, which were not used for refinement.

phospholipids. The presence of two diacyl lipids suggests that the protein could also be able to bind one tetra-acyl lipid, such as diphosphatidylglycerol (cardiolipin).

To investigate the mechanism of entry and release of the two lipids in the cavity of Ttg2D$_{Pae}$, we performed a normal mode analysis (NMA). NMA may be used to model the internal collective motions of a protein, relevant to ligand biding and function in general, typically described by a few low-frequency modes[37]. Figure 1d shows the collective motions along mode 7, the first non-trivial mode (modes 1–6 account for translational and rotational motions of the protein as a whole). Rather than "en bloc" relative motions of sub-domains, all secondary structures of the protein appear to move in a concerted manner, helix α4 and the core of the β-sheet being more rigid. This breathing-like motion increases in a concerted manner the volume of the cavity and its mouth area, and may allow the lipids to enter into or exit from the cavity. Inspection of the next 10 lowest-frequency normal modes shows similar concerted motions. The recent MalC structure with no lipid bound (PDB entry 6GKI)[32] shows similar collective motions along all modes, with similar amplitudes, indicating that the cavity could open and close in the absence of

lipid. The normal modes can be also used to compute atomic mean-square displacements, which can be in turn related to $B$-factors[38]. The NMA-derived and the observed (crystallographic) $B$-factors are closely correlated except in regions 75–95 and 180–200, which are involved in crystal contacts, and region 105–120, where the electron density is weaker (Supplementary Fig. 3). This suggests that the normal modes provide a realistic description of the protein's flexibility.

**The Ttg2D/MlaC fold: a new two-domain architecture and SBP structural cluster.** All MlaC homologs of known structure have a highly superposable "decanter"-shaped configuration (Fig. 1a), previously described as an "extended" NTF2 fold[27] but never assigned a distinct structural classification. Thus, while the *P. putida* structure (PDB entry 4FCZ) appears in the CATH database as a single domain protein and unique structure of superfamily 3.10.450.710, belonging to fold topology 3.10.450 (Nuclear Transport Factor 2; Chain: A), the *R. solanacearum* structure (2QGU) is described as a two-domain structure with domains belonging to CATH-superfamilies 3.10.450.50 (NTF2-like) and 1.10.10.640. The latter superfamily belongs to the all-alpha fold topology 1.10.10 (Arc Repressor Mutant, subunit A) and has 2QGU as unique structure. To clarify the structural classification of Ttg2D/MlaC proteins we run a DALI search of the putative second domain (D2 in Supplementary Fig. 1) of Ttg2D$_{Pae}$ against the whole PDB. The results showed very good superposition with the small alpha domain of several AAA+ proteins, the best match being with 6UKS chain C[39] (Supplementary Fig. 4). AAA (ATPases Associated with diverse cellular Activities) domains are formed by a large N-terminal domain adopting a Rossmann fold and a small C-terminal domain forming an alpha-helical bundle. They tend to adopt homo-hexameric ring complexes and hydrolyze ATP to perform activities that involve protein remodeling. The helical domain plays an important role in coupling the conformational changes resulting from ATP hydrolysis to the neighbor monomer within the AAA-ring and to the underlying protease ring[40] and has a specific classification (other than AAA) both in the PFAM (PF17862) and CATH databases (1.10.8.60). Interestingly, the Ttg2C ortholog MlaD, which in *E. coli* interacts with MlaC[32], forms also a homo-hexameric ring. We therefore conclude that currently known Ttg2D/MlaC structures combine two structural domains in an architecture not seen yet in other proteins, a NTF2-like domain and an AAA helical-bundle domain. In Ttg2D$_{Pae}$, the first domain is formed by two non-contiguous sequence segments: D1S1 (PDB residues 23–68), with three alpha-helices, and D1S2 (113–169), with five beta-strands. The second domain, with five helices, is also split in two non-contiguous sequence segments: D2S1 (69–112) and D2S2 (170–212) (Fig. 1a and Supplementary Fig. 1).

Classical SBPs linked to ABC transporters are structurally similar and composed of two globular domains formed by discontinuous segments[41], adopting either a Rossmann fold (CATH: 3.40.50) or a very similar CATH: 3.40.190. The 501 SBP structures available in the PDB have been classified into seven clusters and several subclusters[42]. We superposed the Ttg2D$_{Pae}$ structure to a representative of each of these subclusters. RMSD values, number of aligned residues, and structural superpositions are shown in Supplementary Fig. 5. The longest match aligns 47 residues with an RMSD of 3.97 Å (2PRS chain A, a 284-residue member of cluster A–I), while the best RMSD is 1.41 Å with 23 residues aligned (3MQ4 chain A, a 481-residue member of cluster B–V). These results clearly confirm that Ttg2D/MlaC family proteins do not belong to any previously known SBP structural cluster.

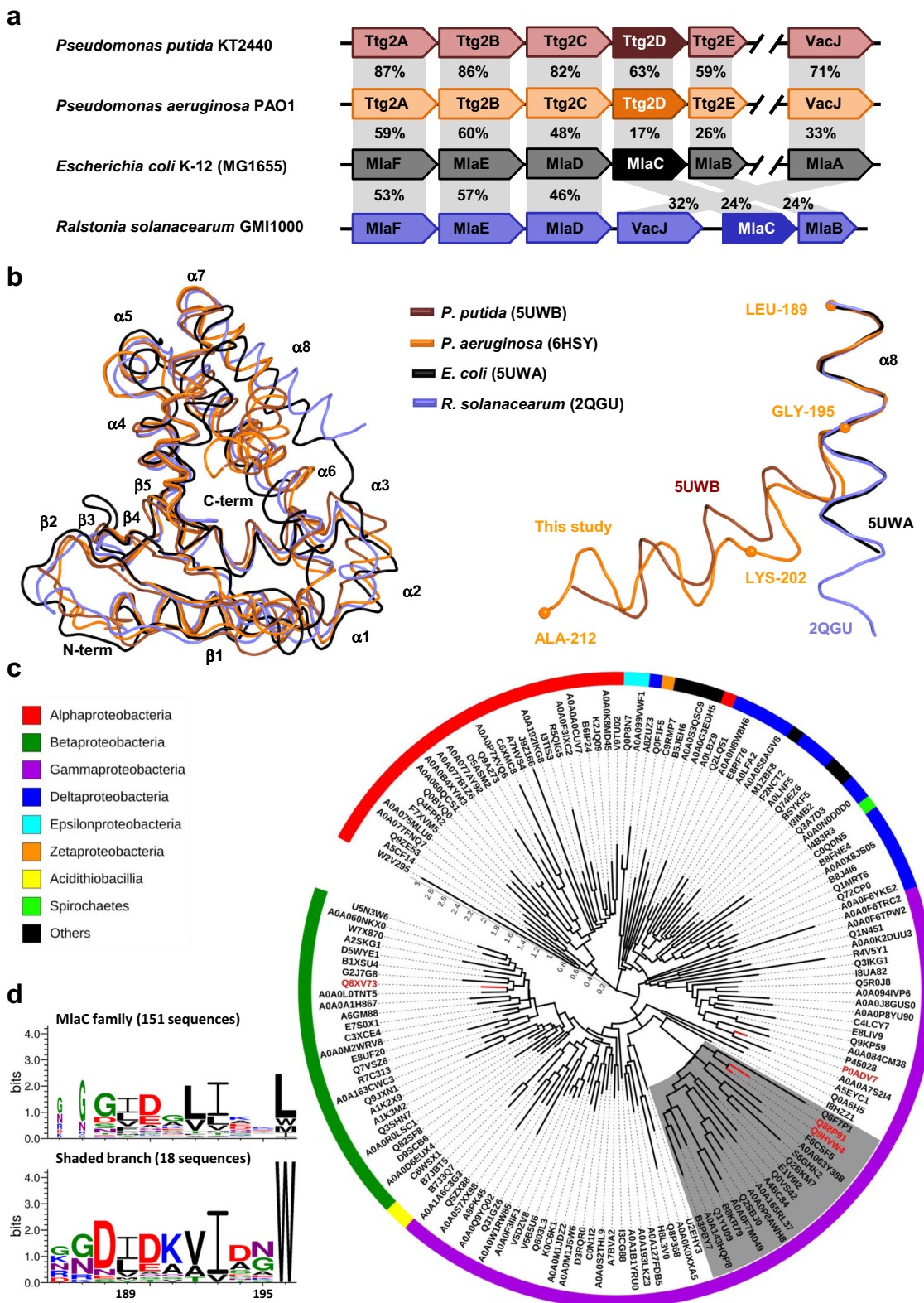

**Evolution of sequence and structural diversity of the MlaC family**. A structural alignment of Ttg2D$_{Pae}$ with other MlaC family proteins with known 3D structures reveals that, despite sequence identities ranging from 63% for the *P. putida* protein to as low as 17% for the *E. coli* one (Fig. 2a, b), the RMSDs of the structural alignments are very low, ranging from 1.6 to 3.1 Å (188–185 Cα), respectively (Supplementary Table 1). Clearly, secondary structure elements are highly or strictly conserved

among all four proteins, despite substantial amino acid variation (Fig. 2b and Supplementary Fig. 1). However, the four proteins split into two groups: *P. aeruginosa* and *P. putida* Ttg2D have a hydrophobic cavity of 2979–2337 Å$^3$ and can bind two diacyl lipids, while the *R. solanacearum* and *E. coli* proteins have a half-size cavity of 1444–1332 Å$^3$ and bind only one diacyl lipid (Supplementary Table 1). Surprisingly, although the different number of ligands had been already noticed when the structure of

**Fig. 2 Sequence and structural diversity among MlaC family proteins. a** Genetic organization of the Ttg2/Mla pathways in different bacterial species based on NCBI reference genomes. The amino acid identities (%) between ortholog pairs as determined by Clustal Omega are shown on gray background. **b** Superposition of the structure of Ttg2D from *P. aeruginosa* with known structures of ortholog proteins (PDB codes indicated). Left panel shows superposition of whole structures and right panel highlights the superposition of helix α8. **c** Phylogeny of 151 representative amino acid sequences belonging to the Pfam family MlaC (PF05494) and identified across different Gram-negative species (bacterial classes are shown with different colors). The maximum likelihood tree was calculated with the LG + G + F model on MEGA 7, and visualized and annotated with iTOL. UniProt codes are used to identify each sequence, and those proteins with known 3D structure are indicated in red (P0ADV7 in *E. coli*, Q8XV73 in *R. solanacearum*, Q88P91 in *P. putida*, and Q9HVW4 in *P. aeruginosa*). The shaded branch corresponds to proteins that we predict to bind two diacyl lipids. This branch comprises sequences from different species belonging to four orders of Gamma-proteobacteria (*Pseudomonadales*, *Alteromonadales*, *Cellvibrionales*, and *Oceanospirillales*). **d** The aligned sequence logo shows a signature motif that distinguishes the sub-set of sequences corresponding to the shaded branch of the tree (18 sequences) from the whole set of 151 representative sequences of the MlaC family. At a given position, the height of a residue is proportional to its frequency. Residue numbering is according to the sequence in *P. aeruginosa*. Source data and the sequence alignment are provided as Supplementary Data 1.

Ttg2D from *P. putida* was solved, cavity differences were never analyzed. Figure 1b illustrates the cavity difference between *P. aeruginosa* and *R. solanacearum* Ttg2D. The volume differences correlate with the different number of residues forming the cavities, from 55 down to 31 (Supplementary Table 1). However, these residues, which are spread along the whole protein sequence (Supplementary Fig. 1), are largely conserved in terms of position and, in most cases, in terms of identity or similarity, with a few substitutions such as V147/L, or V163/I or M directly affecting the volume. Some side-chain reorientations, like Y105, and small secondary structure displacements, like strands β3 and β4 or helix α6 shifted by ~2 Å (Fig. 2b and Supplementary Fig. 6), also modulate the volume. Taken together these changes are, nevertheless, not sufficient to explain how the cavity volume can double. Helix α8 seems to be the main responsible for the difference between a two and a one diacyl-phospholipid cavity, not only because the helix is longer in the first case (Supplementary Fig. 1) but also because it adopts a different conformation. Indeed, for the second group (*R. solanacearum* and *E. coli*, one diacyl lipid), this helix has a straight conformation, covers the α6 helix (Fig. 2b and Supplementary Fig. 6) and does not participate in the cavity (Fig. 1b and Supplementary Fig. 1), while in the first group (*P. aeruginosa* and *P. putida*, two diacyl lipids), the α8 helix is bent towards and over the α7 helix and greatly enlarges the cavity (with additional residues from β4 and β5 strands). This bend occurs at residue G195 with an angle of 40° and 64° in Ttg2D proteins from *P. aeruginosa* and *P. putida*, respectively (Fig. 2b). The helix of the first protein has an additional bend of 43° at K202. Glycine has a poor helix-forming propensity[43] and tends to disrupt helices because of its high conformational flexibility. On the other hand, the Phe197 and Gln196 residues occupying the Gly195 position in the *R. solanacearum* and *E. coli* proteins, respectively (Supplementary Fig. 1), have better helix-forming propensities and maintain the α8 helix straight. In addition, W196, exclusive of the pseudomonal structures, may also contribute to the influence of the α8 helix on the cavity's volume, since its bulky hydrophobic side chain, deeply inserted into a hydrophobic pocket on the concave side of the curvature, could stabilize the helix α8 bend (Supplementary Fig. 6).

Components of the Mla system are broadly conserved in Gram-negative bacteria, except for the periplasmic MlaC that notoriously shows high inter-species sequence diversity (Fig. 2c). Interestingly, an alignment of 151 representative amino acid sequences belonging to the MlaC family and identified across different Gram-negative species revealed that W196 is conserved not only in *Pseudomonas* species but also in a group of related sequences in other non-phylogenetically related gamma-proteobacteria (Fig. 2d). In this group of proteins that would hypothetically bind two diacyl phospholipids, other positions with distinct residues relative to other MlaC family members

stand out, especially in two regions located between the central part and the C-terminal end of the protein (Supplementary Fig. 6). Side-chain orientation and hydrophobicity of some residues in these regions could be also contributing to a tighter binding of the two diacyl phospholipids inside the ligand cavity. The presence of common protein sequence signatures in species that are not closely related indicates that horizontal gene transfer, mediated by recombination events between flanking conserved genes, could have contributed to MlaC family diversity.

**Ttg2D$_{Pae}$ binds two diacyl glycerophospholipids and cardiolipin, to our knowledge representing a novel phospholipid trafficking mechanism among Gram-negative bacteria.** Native MS was used to determine the lipids that bind specifically to recombinant Ttg2D$_{Pae}$ produced in the cytoplasm of *E. coli* and the stoichiometry of the interaction in a cellular environment (Fig. 3). The native mass spectrum of Ttg2D$_{Pae}$ shows two major charge states ($m/z$ 2701, $z = 9$ and $m/z$ 2430, $z = 10$) corresponding to a deconvoluted average mass of 24,296 Da that matches the MW of the recombinant protein Ttg2D$_{Pae}$ produced in *E. coli* without the first methionine residue plus bound ligands (Fig. 3a and Supplementary Fig. 7). After isolation of the wide peak ion at $m/z$ 2700 ($z = 9$) and gas-phase fragmentation with a transfer collision energy (CE) of 50 V, we detected the unbound protein ($m/z$ 2536, $z = 9$) and a family of released phospholipids in the low mass range (Fig. 3b, c). Additional native MS experiments on isolated ion $m/z$ 2700 ($z = 9$) using increasing transfer CE confirmed that at least a fraction of the bound Ttg2D$_{Pae}$ population hosts two phospholipids, since lipid dissociation starts at a transfer CE of 35 V and goes through the single-bound species (Supplementary Fig. 8a). The results also show that a second fraction of the Ttg2D population hosts a single molecule that remains bound up to a transfer CE of 50 V, which could correspond to cardiolipin as further MS experiments indicated. A tighter binding of cardiolipins could explain why these molecules were not detected in the gas-phase fragmentation experiments (Fig. 3b, c). We also note that at the transfer CE required for cardiolipin release these molecules are easily fragmented (see below). Fragmentation of the isolated peak $m/z$ 2430 ($z = 10$) showed similar results (Supplementary Fig. 8b).

The mass of [M+H]+ ions at $m/z$ 664.5, 704.5, 718.5, and 730.5 released from Ttg2D$_{Pae}$ confirmed the identity of some ligands as phosphatidylethanolamines (PE) with different hydrocarbon chains (PE C30:0, PE C33:1, PE C34:1, and PE C35:2, respectively) (Fig. 3c). Subsequent direct MS analysis under denaturing conditions in both positive and negative ion modes identified additional species of PE and of other phospholipid classes that are also components of the bacterial membrane, such as phosphatidylglycerol (PG), phosphatidylcholine, and phosphatidylserine (Fig. 3d and Supplementary Fig. 9). With these

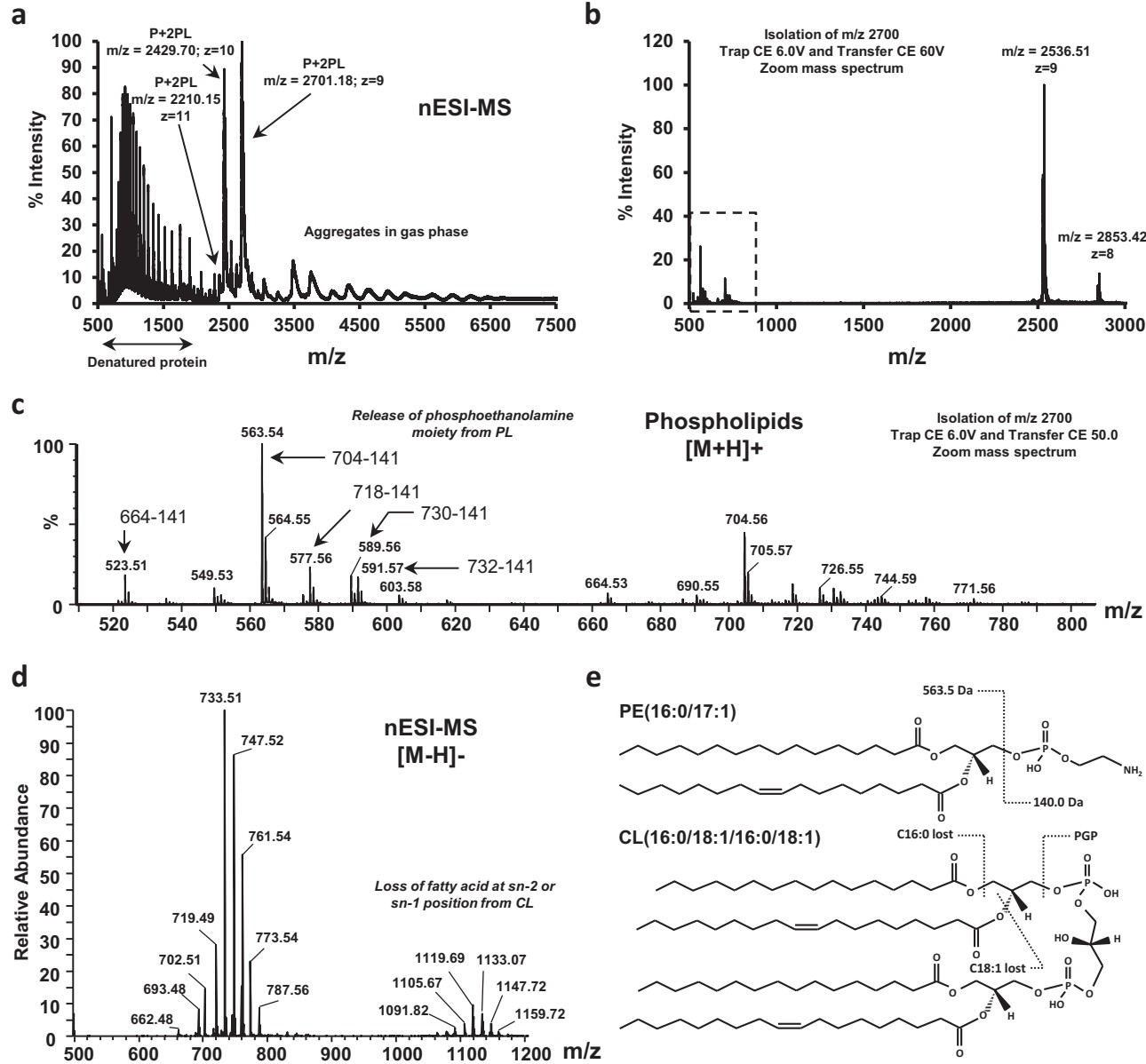

**Fig. 3 Ttg2D_Pae is a promiscuous phospholipid-binding protein. a** Mass spectrum of Ttg2D_Pae (produced in *E. coli*) under native conditions (native MS), nanoelectrospray ionization (nESI) in positive ion mode. The two major charge states ($z = 9$ and $z = 10$) correspond to a deconvoluted average mass that matches the MW of the recombinant protein (P) (22836.78 Da) plus two phospholipid (2PL) molecules. The peaks $m/z$ 500–1500 correspond to partially denatured proteins and the peaks $m/z$ 3000–6000 correlate with protein aggregates formed in gas phase. **b** Fragmentation mass spectrum for the isolated ion at $m/z = 2700$ ($z = 9$); PLs are expected in the range 600–800 Da. **c** Zoom of the MS/MS spectrum in positive mode (boxed in panel **b**) showing the most abundant glycerophospholipids released from Ttg2D_Pae under non-denaturing MS dissociation conditions. The major PLs detected show a loss of 141 Da, after the MS/MS experiment, that corresponds to the release of a phosphoethanolamine (PE) moiety. **d** Negative ion mode mass spectrum under denaturing conditions of glycerophospholipids released by Ttg2D_Pae (FT resolution 100k and accuracy <3 ppm). With this ionization mode, phosphatidylglycerol (PG) species are mostly detected. A detailed list of detected peaks is shown in Supplementary Table 2. We propose that peaks $m/z$ 1050–1160 correspond to *E. coli* cardiolipin species that have lost one of the fatty acids at *sn*-1 or *sn*-2. **e** Structure of the most common species of PE and cardiolipin in *E. coli* showing the most favorable fragmentation sites. PGP phosphatidylglycerol-phosphate.

methods, PG C33:1, PG C34:1, PE C33:1, and PE C35:1 came out as most abundant (annotation of lipid species based on the most abundant fatty acids in *E. coli* is given in Supplementary Table 2). The observed distribution of phospholipids bound to recombinant Ttg2D_Pae relates to their relative abundances in *E. coli* BL21 (the recombinant protein-expression host) as determined by lipidome analyses (Supplementary Fig. 10), and it correlates well with the reported phospholipid composition of *E. coli* under comparable conditions[44,45].

LC-MS analysis under denaturing conditions in positive ion mode of the complexes produced in *E. coli* shows lipid species of $m/z$ 690–800 and $m/z$ 1400–1500 that could correspond to phospholipids and cardiolipins, respectively (Supplementary Fig. 9d). In addition, direct MS analysis in negative ion mode indicates the presence of glycerophospholipids and shows peaks at $m/z$ 1050–1160 that could correspond to *E. coli* cardiolipin species having lost one fatty acid (Fig. 3d, e). During the fragmentation processes of phospholipids in negative ion mode,

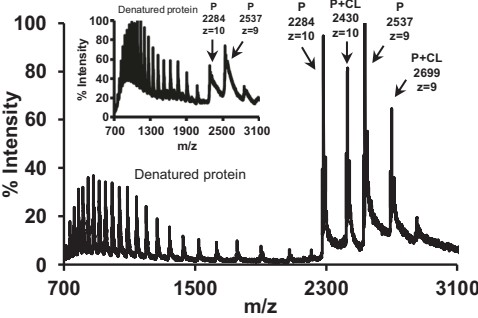

**Fig. 4 Ttg2D_Pae binds cardiolipin in vitro.** Mass spectrum under native conditions in positive ion mode of delipidated protein Ttg2D_Pae (P) produced in *E. coli* (MW 22836.78 Da) after incubation with free cardiolipin CL(18:0)$_4$ (MW 1466.06 Da) at a ratio 1:9 (P:CL). The inset shows the ion chromatogram of the delipidated Ttg2D_Pae sample before incubation.

upon low energy collisional activation, ions corresponding to the loss of fatty acids are the most abundant[46]. For example, when the most abundant cardiolipin species in *E. coli*, i.e. CL C68:2 with MW 1405.0 Da (Supplementary Fig. 10), loses one of the fatty acids at position *sn*-2, it shows a prominent fragment ion at *m/z* 1147 or 1121 depending on the fatty acid species in that position (Fig. 3d, e).

In light of these results, we performed additional native MS experiments to investigate whether delipidated Ttg2D_Pae can bind cardiolipin in vitro. In Fig. 4, representative native mass spectra of a reaction mixture containing 26.25 µM delipidated protein and 245 µM cardiolipin CL(18:0)$_4$ (1:9 molar ratio) in a buffered aqueous solution, are shown. The protein–cardiolipin complex and the unbound protein were detected around the 10+ and 9+ charge states (Fig. 4 and Supplementary Fig. 11). Notably, the dissociation energy needed to release cardiolipin was higher than that required to release diacyl glycerophospholipids in the native complex (transfer CE of 60 vs 35 V, see Supplementary Figs. 11a and 8, respectively). This correlates with the previous observation that a "P + 2PL" population in the native spectra in Supplementary Fig. 8 loses its cargo only at 60 V. At the transfer CE required for cardiolipin release this molecule is in fact easily fragmented (Supplementary Fig. 11b).

**Ttg2D_Pae binds two diacyl glycerophospholipids or cardiolipin in the periplasm of *P. aeruginosa*.** Given that the cytoplasm of *E. coli* is clearly not the natural environment of Ttg2D_Pae, we decided to produce and purify this protein directly from the periplasmic space of *P. aeruginosa*. For this purpose, protein Ttg2D_Pae tagged with six C-terminal histidine residues and containing its own N-terminal secretion signal was expressed in the genetic background of a *P. aeruginosa* PAO1 Δttg2D mutant using a derivative of a broad-host-range cloning vector (Fig. 5a). Restoration of the colistin susceptibility phenotype after transformation (minimum inhibitory concentration (MIC) of 0.25 vs. 0.0625 µg/ml in the non-transformed mutant) was used as evidence that the protein being expressed in the mutant strain was functional. Under these conditions, the mature Ttg2D_Pae could be purified from the periplasm in sufficient amount and purity (Fig. 5b, inset) to determine its phospholipid cargo by MS.

The native mass spectrum in Fig. 5b shows a charge-state distribution that corresponds to the ligand-bound mature Ttg2D_Pae complexes (MW of intact complex: 23621 Da; MW of His-tagged protein without the signal peptide: 22,140 Da). The wide peak widths, besides poor desolvatation in aqueous buffer in the native MS instrumental conditions, suggests also the coexistence of multiple species of similar mass, supporting

the presence of different classes of phospholipids bound to the protein. Gas-phase dissociation mass spectra at varying transfer CE of isolated ions around *m/z* = 2625 (*z* = 9) indicated the presence of two populations at peaks *m/z* = 2626 (*z* = 9) and *m/z* = 2619 (*z* = 9), which are in agreement with the binding of two phospholipids and one cardiolipin, respectively (Fig. 5c). Further MS experiments under denaturing conditions in negative ion mode allowed the identification of two main phospholipid classes (Supplementary Fig. 12 and Supplementary Table 3). The obtained distribution is in agreement with relative abundances in *P. aeruginosa* PAO1 as determined by a lipidome analysis (Supplementary Fig. 10), and it correlates well with the reported phospholipid composition in this species[47].

**The Ttg2 system provides *P. aeruginosa* with a mechanism of resistance to membrane-damaging agents.** As expected, the *P. aeruginosa* Δttg2D mutant exhibited a debilitated outer membrane leading to increased susceptibility to several membrane-damaging agents (Fig. 6), as demonstrated by the 1-*N*-phenylnaphthylamine (NPN) assay. Indeed, an enhancement in NPN uptake was observed in the mutant in the presence of the permeabilizer agents EDTA and colistin (Fig. 6a, b). In line with this, the Δttg2D mutant is more susceptible to the action of polymyxins (lipid-mediated uptake), but also of antibiotics that use both the lipid- and porin-mediated pathways to penetrate the cell, including fluoroquinolones, tetracyclines, and chloramphenicol (Fig. 6c). With regard to polymyxin antibiotics, the *ttg2D* transposon insertion mutant was eightfold more susceptible to colistin than the PAO1 wild type, a colistin-susceptible reference strain (Supplementary Table 4). In general, the mutation did not significantly affect the resistance phenotype displayed by the PAO1 strain to the beta-lactam antibiotics or aminoglycosides tested. The susceptibility phenotypes due to deletion of *ttg2D* could be fully or partially reverted by complementation with the cloned *ttg2D* gene or the full operon *ttg2* in the replicative broad-range vector pBBR1MCS-5 (Fig. 6 and Supplementary Table 4), confirming the link between the gene and the phenotypes. We have also confirmed that insertional mutations in each of the other components of the *ttg2* operon (*ttg2A*, *ttg2B*, *ttg2C*) and *vacJ* (*mlaA* ortholog) lead to an increased susceptibility to antibiotics in the same way as for the Δttg2D mutant (Supplementary Table 4). The Δttg2D mutant is also considerably susceptible to the toxic effect of the organic solvent xylene (Fig. 6d) and it is fourfold more susceptible to the chelating agent EDTA (MIC = 0.5 mM) than the parental wild-type PAO1. However, no difference was observed between the mutant and wild-type cells in their susceptibility to SDS, obtaining for both strains a MIC value of 0.8%. Finally, disruption of the *ttg2D* gene resulted in an approximately twofold reduction in biofilm formation and greatly increased the activity of EDTA against *P. aeruginosa* biofilms at a subinhibitory concentration of 0.05 mM (Fig. 6e).

**The Ttg2 system is associated to *P. aeruginosa*'s intrinsic resistance to low antibiotic concentrations.** The susceptibility of Ttg2-defective mutants to antibiotics was further studied in strains with different genetic backgrounds. To this end, the full *ttg2* operon was mutated in the clinical MDR *P. aeruginosa* strains C17, PAER-10821, and LESB58, which had shown different patterns of resistance to several antibiotic classes, specifically, polymyxins, fluoroquinolones, and tetracyclines (Table 2). In particular, PAER-10821 and LESB58 are *P. aeruginosa* strains with low-level resistance to colistin. The generation of mutants with disrupted gene functions in MDR bacteria is troublesome because the antibiotics commonly used in the laboratory are no longer useful for selection of gene knockouts. In addition, the loci

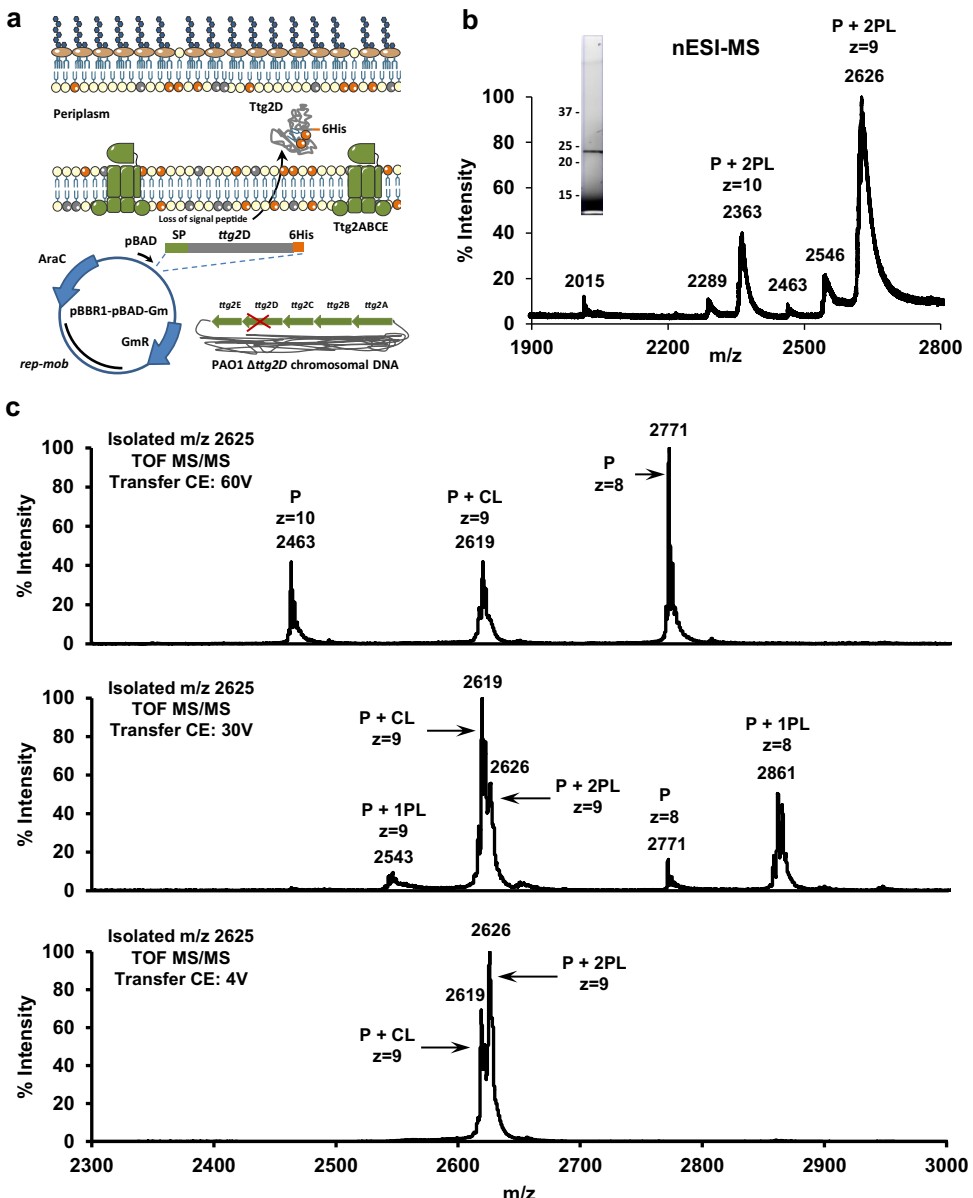

**Fig. 5 Phospholipids carried by Ttg2D$_{Pae}$ in the periplasm of *P. aeruginosa*. a** Schematic representation of the production of homologous protein Ttg2D$_{Pae}$ with a 6xHis in the periplasm of *a P. aeruginosa* PAO1 mutant strain lacking the wild-type Ttg2D component of the Ttg2 system. A pBBR1MCS derivative plasmid was used for the expression of the *ttg2D* gene under the control of the arabinose promoter (pBAD) and containing its own signal peptide (SP). Plasmid construction and its genetic elements are described in the Supplementary Methods. **b** Positive ion mode, non-denaturing mass spectrum of protein Ttg2D$_{Pae}$ produced in *P. aeruginosa*. The His-tagged protein has a molecular weight of 22140.37 Da after cleavage of the predicted signal peptide. Purified protein is shown in the inset. **c** Fragment ion mass spectra of isolated ions around $m/z = 2626$ ($z = 9$) using varying transfer collision energies (CE). These results confirm that a fraction of the Ttg2D$_{Pae}$ (P) population purified from the periplasmic space of *P. aeruginosa* is complexed with two phospholipids (PL) and another fraction binds cardiolipins (CL). The most abundant cardiolipin species in strain PAO1 based on a lipidomic analysis (Supplementary Fig. 10b) are CL(68:2) with MW 1405.0 Da and CL(70:5) with MW 1427.0.

mutated in this case is involved in a general mechanism of resistance to antimicrobial agents and mutant strains are therefore expected to be generally susceptible and thus potentially lost during the selection steps. For this reason we have adapted a mutagenesis system based on the homing endonuclease I-SceI[48,49] to construct targeted, non-polar, unmarked gene deletions in MDR *P. aeruginosa* strains (see "Methods" and Supplementary Fig. 13 for details). With this modified mutagenesis strategy we have obtained and validated unmarked deletion mutants of the selected MDR strains lacking the full *ttg2* operon (Supplementary Fig. 13). Complemented strains were also obtained by transformation of mutant strains with a replicative

plasmid containing the full *ttg2* operon and its expression in the complemented clones was confirmed by reverse transcription PCR (RT-PCR) (Supplementary Fig. 13). All these strains were tested for their susceptibilities to different classes of antibiotics (Table 2).

The three *ttg2* mutants were considerably more susceptible (between 4- and 64-fold) than the corresponding wild-type bacteria to colistin, fluoroquinolones, or tetracycline analogs, but not to the other antibiotic classes (Table 2). The mutant susceptibility phenotypes could be reverted by providing an intact copy of the entire PAO1 *ttg2* operon (PA4456–PA4452) in a replicative plasmid, except for colistin. The lack of

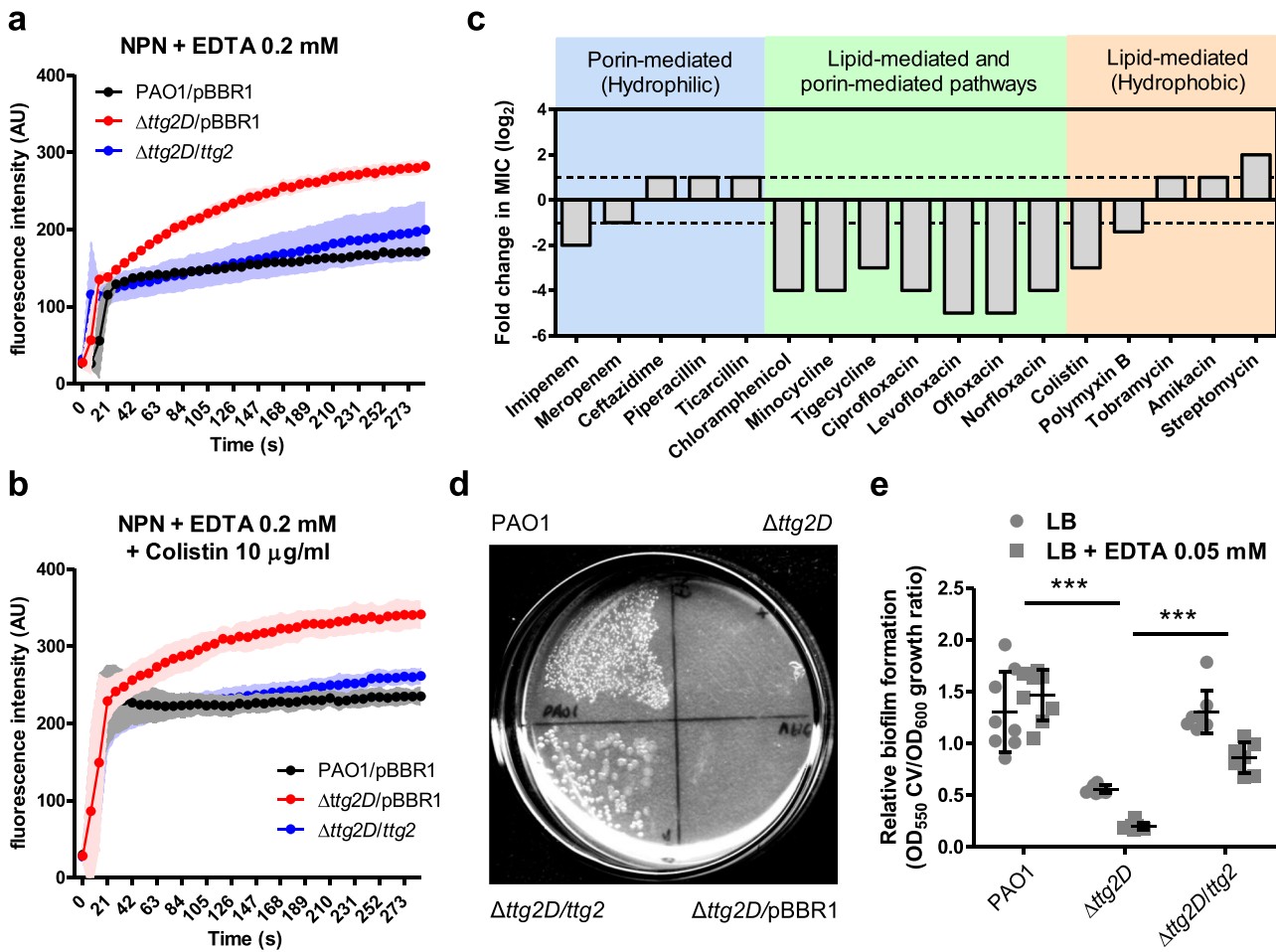

**Fig. 6 Phenotypic changes in the Δttg2D P. aeruginosa mutant denote destabilization of its outer membrane. a, b** Ability of EDTA and colistin to permeabilize the outer membrane (NPN assay) of the native, mutant, and complemented PAO1. Data presented are the mean ± SD from three independent experiments. **c** Relative change of the Δttg2D mutant MIC (minimum inhibitory concentration) for antibiotics of different classes grouped according to their cell entry mechanism. Fold changes were determined with respect to the PAO1 wild type, represented as dotted lines (source data in Supplementary Table 4). **d** Growth in LBMg plates overlaid with 100% p-xylene. Under this condition the growth was assessed following incubation at 37 °C for 24 h. The image is representative of duplicate experiments. **e** Relative biofilm formation determined by crystal violet (CV) staining for Δttg2D mutant and control strains in LB medium with and without EDTA. Error bars represent SD of the mean ($n = 8$). Asterisks denote the significance of the data between groups (one-way ANOVA with Tukey's multiple comparison test). In all panels pBBR1 indicates insertion of pBBR1MCS-5 vector alone as a control. Source data are provided as Supplementary Data 1 for **a, b, d** and **e**.

complementation of the colistin susceptibility phenotype could be due to the effect of the antibiotic erythromycin (used as a selection marker for complemented strains) on the expression of global regulators that may influence colistin susceptibility[50,51] or to the overexpression of the *ttg2* operon components (two- to eightfold with respect to wild type, see Supplementary Fig. 13) that may also affect the distribution of phospholipids in the OM. Surprisingly, the susceptibility to amikacin decreased for the C17 mutant and an opposite effect was observed for the LESB58 mutant and tobramycin, suggesting a genetic-background component in the effect of the *ttg2* mutation on the susceptibility to these antibiotics.

## Discussion

We have performed a structural and functional study of the soluble periplasmic SBP component of the Ttg2 ABC transport system in *P. aeruginosa* (Ttg2D$_{Pae}$) that, to our knowledge, reveals new facets of this protein family and provides additional insight into the role of this pathway in *P. aeruginosa*. We have first characterized this protein at the molecular level, supporting its predicted role as a phospholipid transporter. The crystal structure

of recombinant Ttg2D$_{Pae}$ (Fig. 1) and MS analysis of protein–ligand complexes formed in vivo and in vitro (Figs. 3–5) show that this SBP transports four acyl chains. Furthermore, our results demonstrate that the Ttg2 system in *P. aeruginosa* is a general glycerophospholipid transporter with the ability to carry either two phospholipids or a tetra-acyl cardiolipin-like species. Reevaluation of the Ttg2D structure from *P. putida* (PDB 4FCZ) by Ekiert et al.[27] (PDB 5UWB) had previously suggested but not confirmed the presence of a tetra-acyl, cardiolipin-like lipid in its bulky hydrophobic pocket. This is in contrast with the Ttg2D/ MlaC orthologs from *E. coli* (PDB 5UWA) and *R. solanacearum* (PDB 2QGU), which seem to bind a single diacylglyceride based on electron density and cavity size. Although, more recently, Hughes et al.[32] showed evidences of cardiolipin binding to *E. coli* MlaC, these have yet to be confirmed, since the methodology used does not allow the distinction between specific binding and co-purification (unspecific). With the data at hand, it thus appears that, among Gram-negative bacteria, the ability of MlaC-family proteins to transport two phospholipids or cardiolipin is exclusive to some taxonomic groups. Phylogenetic and sequence analysis (Fig. 2) suggest that there are other genera in addition to

**Table 2 Antibiotic susceptibility profile of *P. aeruginosa* MDR strains lacking the full *ttg2* operon.**

| Antibiotic | MIC$^a$ in µg/ml | | | | | |
| --- | --- | --- | --- | --- | --- | --- |
| | LESB58 | | C17 | | PAER-10821 | |
| | WT | Δ*ttg2* | WT | Δ*ttg2* | WT | Δ*ttg2* |
| Polypeptides | | | | | | |
| Colistin | 4 | 0.125* | 2 | 0.125* | 32 | 32 |
| Fluoroquinolones | | | | | | |
| Ciprofloxacin | 2 | 1 | 256 | 64* | 256 | 128 |
| Levofloxacin | 8 | 2* | 256 | 256 | 256 | 128 |
| Ofloxacin | 16 | 4* | >32 | >32 | >32 | >32 |
| Norfloxacin | 8 | 4 | >256 | >256 | >256 | 256 |
| Tetracyclines | | | | | | |
| Tetracycline | 16 | 8 | 32 | 16 | 32 | 8* |
| Minocycline | 32 | 8* | 16 | 8 | 32 | 8* |
| Tigecycline | 16 | 8 | 64 | 8* | 32 | 8* |
| Chloramphenicol | | | | | | |
| Chloramphenicol | 32 | 32 | 128 | 64 | 128 | 64 |
| Sulfonamides | | | | | | |
| Trimethoprim-sulfamethoxazole | 16 | 8 | >64 | >64 | >64 | 64 |
| Aminoglycosides | | | | | | |
| Tobramycin | 8 | 2* | 64 | 128 | 128 | >128 |
| Amikacin | 64 | 64 | 8 | 32* | 32 | 32 |
| Gentamicin | 32 | 16 | >128 | >128 | >128 | >128 |
| Kanamycin | >512 | >512 | 256 | 512 | 512 | 512 |
| Streptomycin | >64 | >64 | >64 | >64 | >64 | >64 |
| Carbapenems (beta-lactam) | | | | | | |
| Imipenem | 2 | 2 | 32 | 32 | 32 | 64 |
| Meropenem | 2 | 2 | 32 | 16 | 16 | 16 |
| Cephalosporins (beta-lactam) | | | | | | |
| Ceftazidime | 256 | 256 | 64 | 128 | 16 | 32 |
| Penicillins (beta-lactam) | | | | | | |
| Piperacillin | 256 | 256 | 256 | >256 | 128 | 256 |
| Piperacillin–tazobactam | 128 | 128 | 256 | >256 | 64 | 64 |
| Ticarcillin | >256 | >256 | >256 | 256 | 64 | 128 |
| Ticarcillin–clavulanic acid | >32 | >32 | >32 | >32 | >32 | >32 |

*MIC differences greater than twofold with respect to the corresponding wild-type strain were considered significant.
$^a$Minimum inhibitory concentration (MIC) determined by the broth microdilution method. MICs were confirmed by two or three independent replicates.

*Pseudomonas* where the Mla system transports two molecules simultaneously, although this needs to be confirmed by further studies. This finding raises the question whether the evolution of this system in these species has been driven by transport efficiency (double cargo) or transport diversity (tetra-acyl in addition to diacyl phospholipids). Furthermore, are the two phospholipids translocated simultaneously by the permease Ttg2B, as it would need to be for a tetra-acyl phospholipid such as cardiolipin? The determination of the structure of additional transport components in other species will be necessary to corroborate our proposed classification and answer these questions.

MS analyses showed that Ttg2D$_{Pae}$ is a highly promiscuous SBP. This protein is indeed able to bind in vivo phospholipids with different head groups, particularly PG and PE, and of different chain lengths and degree of unsaturation. Therefore, this system would not only control the global phospholipid content of the OM, but could be also controlling a precise membrane-lipid distribution. Bacterial cells tightly regulate the phospholipid composition of the OM to fortify the permeability barrier against small toxic molecules, including antibiotics. For example, anionic phospholipids like PG interact with membrane proteins and cationic antimicrobials in ways that zwitterionic phospholipids like PE do not, their balance requiring a fine control[19,52]. Indeed, the membrane's PE content is a major factor determining the bacterial susceptibility to certain antimicrobial agents[52,53]. On the other hand, anionic non-bilayer-forming phospholipids like

cardiolipin can strengthen the OM of Gram-negative bacteria against certain antibiotics[19]. In Gram-negative bacteria, cardiolipin is mostly located within the IM (the site of its synthesis), but it is also present in the OM where it facilitates proper localization of proteins on the bacterial surface. Known mechanisms of anterograde transport ensure the presence of cardiolipin in the OM[54], but a retrograde mechanism preventing accumulation in the OM has not been described yet. An increase in the cardiolipin content of the OM could cause an increase in susceptibility to cationic antimicrobial agents[55]. Positively charged antimicrobial peptides and polymyxins have been proposed to promote the clustering of anionic phospholipids leading to phase-boundary defects that transiently breach the permeability barrier of the cell membrane[52]. Beside this, it has recently been shown that penetration of polymyxin B is promoted by increasing membrane surface charge[56]. In *P. aeruginosa*, an organism showing significant intrinsic resistance to certain antibiotics, the membrane PE:PG:cardiolipin composition is approximately 60%:20%:11%[18]. Simultaneous transport of four acyl chains across cell membranes could help control membrane charge balance more efficiently, not only by number but also by inclusion of additional (cardiolipin) species.

Early studies[20] (and more recent ones[57]) of the Ttg2/Mla system supported its role in the maintenance of lipid asymmetry in the Gram-negative OM, by retrograde trafficking of phospholipids from the OM to the cytoplasm through the IM. Conversely, other studies have recently shown that the *E. coli* protein MlaD spontaneously

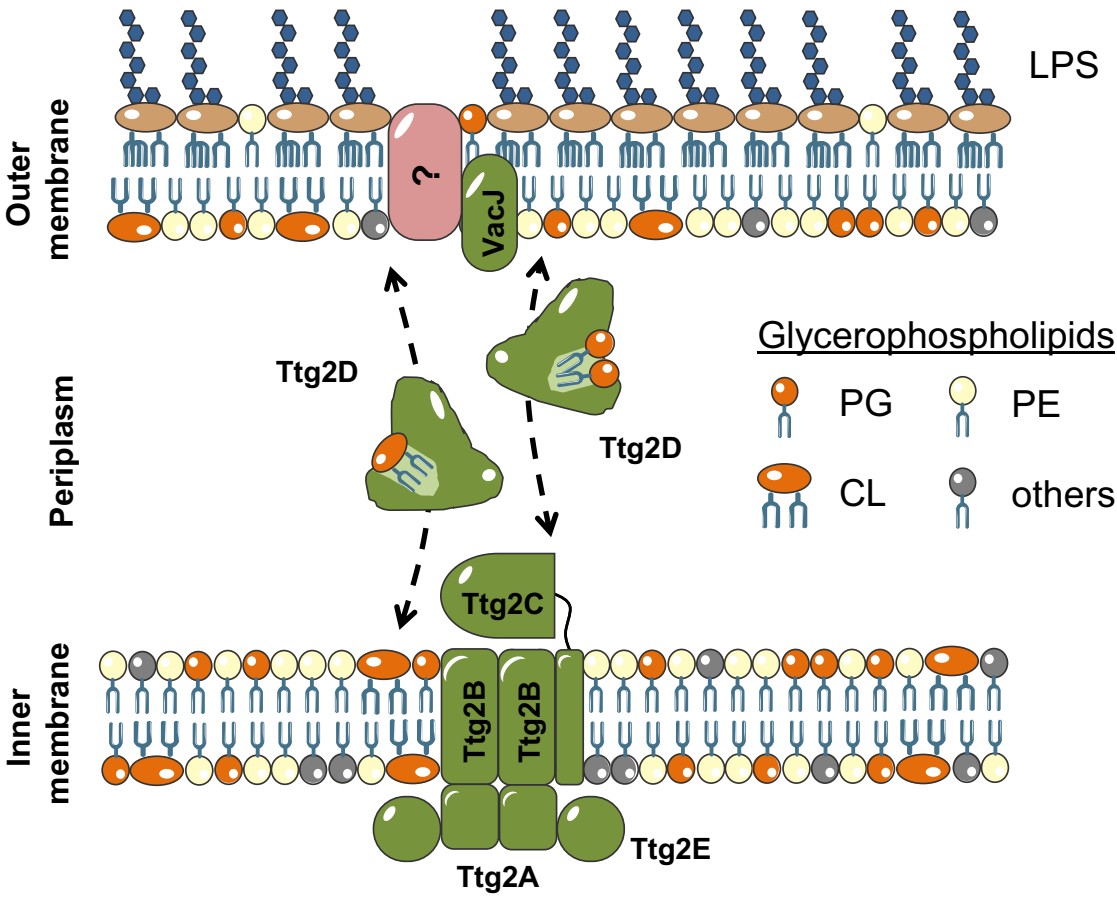

**Fig. 7 Proposed model of the Ttg2 system in *P. aeruginosa*.** The soluble substrate-binding protein Ttg2D$_{Pae}$, together with the inner membrane complex Ttg2ABCE, transports mislocalised glycerophospholipids between the two cell membranes through the periplasm of *P. aeruginosa*. It is not yet known if the VacJ component of this system, which is thought to deliver the phospholipids to Ttg2D$_{Pae}$, forms a complex with specific porins, as in *E. coli*, to extract the lipids from the outer membrane. Ttg2D$_{Pae}$ can carry two glycerophospholipids (e.g phosphoethanolamine PE or phosphatidylglycerol PG) and, based on structural and mass spectrometry data, it could accommodate a tetra-acyl phospholipid such as cardiolipin (CL). Considering recent studies[30, 32], the structural signatures of the ATPase and permease models from the ortholog Mla system in *E. coli*[27], characteristic of importer and exporter ABC cassettes, we propose, in addition to a role in retrograde phospholipid trafficking, a second potential mode of action as an anterograde trafficking system (dashed lines) that would contribute to the maintenance of phospholipid distribution asymmetry. The direction in which this system transports the different phospholipid classes remains to be explored.

transfers phospholipids to MlaC in vitro[32,57], suggesting a transport in the opposite direction (IM-to-OM). In addition, an Mla mutant in *A. baumannii* showed decreased abundance of OM phospholipids and accumulation of newly synthesized phospholipids in the IM[30]. Unlike most of the ABC exporters, ABC importers in Gram-negative bacteria require periplasmic SBPs that provide specificity and high affinity. In addition, it is widely accepted that the direction of substrate transport of ABC transporters can be predicted on the basis of both the sequence of the nucleotide-binding component (ATPase)[58,59] and the transmembrane-domain fold of the permease component[60]. The close orthologs in *E. coli* and *Mycobacterium tuberculosis* of the *P. aeruginosa* ATPase Ttg2A, MlaF, (60% identity) and the Mce protein Mkl (40% identity), respectively, have sequence signatures typically found in prokaryotic ABC import cassettes[20,59]. Moreover, the remote homolog TGD3 from *Arabidopsis thaliana* is also a component of an ABC transport system (TGD) that imports phosphatidic acid to the chloroplasts through its outer and inner envelopes[61]. On the other hand, and although operon *ttg2* resembles the classic organization of an ABC importer, there are evidences at the amino acid sequence level supporting evolution toward anterograde function. For instance, structural similarity searches with Dali (see Supplementary Methods) for the Ttg2B ortholog protein in *A. baumannii* (MlaE, PDB 6IC4 chains G

and H)[30] identified as best match a structure of the human ABCA1 (PDB 5XJY), a known ATP-binding cassette cholesterol/phospholipid exporter[62]. All things considered, we should not rule out the possibility of anterograde, in addition to retrograde, phospholipid trafficking by the Ttg2/Mla system. One possibility would be that of a countercurrent model[63], in which different types of phospholipids would exchange between the two membranes obeying to a gradient (Fig. 7). A countercurrent model would explain how asymmetries in membrane lipid distribution might be achieved by the Ttg2/Mla system, but the direction in which each phospholipid is transported would need to be determined. In the anterograde transport proposed for *E. coli*[32], MlaC is loaded by MlaD, suggesting that, for this to be efficient, MlaC should not bind free phospholipids. Taken back to the *P. aeruginosa* system, we have shown that Ttg2D$_{Pae}$ can load free cardiolipin in vitro, but have no data for the spontaneous binding of two diacyl phospholipids. In this system, the idea that the loading of two molecules may require a helper protein (the MlaD ortholog Ttg2C) seems plausible. This would facilitate a putative mechanism in which the Ttg2 system would contribute to the retrograde transport of single lipids (cardiolipin, maybe diacyl lipids also) as well as to the export, enabled by Ttg2C, of pairs of diacyl lipids to the OM (Fig. 7). The exact function of VacJ (MlaA equivalent) and its relation with the transport in either direction is

still to be determined. While *P. aeruginosa*'s *vacJ* gene is located outside the *ttg2* operon, we have data demonstrating that strains lacking this gene share the same phenotype shown by *ttg2* mutants. In *E. coli*, MlaA forms an active complex with the outer membrane proteins OmpC and OmpF[22,24,64]. However, in *P. aeruginosa* there are no clear orthologs to either of these porins, increasing the singular characteristics of this system in this species and suggesting potential mechanistic differences with the more studied *E. coli* transporter (Fig. 7).

We have provided additional evidence, based on NPN uptake and antimicrobial susceptibility assays, that the Ttg2 system controls the permeability of the OM in *P. aeruginosa* regardless of genetic background. Cellular studies showed that deletion of Ttg2D specifically increases the susceptibility to polymyxin, fluoroquinolone, chloramphenicol, and tetracycline antibiotics in the PAO1 reference strain and in three MDR clinical strains (Fig. 6 and Table 2). This mutant phenotype was observed both in the presence and absence of specific resistance mechanisms providing high-level resistance. For example, PAO1 is a relatively susceptible strain and LESB58 is a MDR strain, and both show diversity in their resistomes[65]. Thus, for the strain and antibiotic panel considered, the increase in susceptibility upon *ttg2* deletion seems to correlate with the antibiotic class rather than with the genetic background. This is in line with the physico-chemical properties of these antimicrobial compounds. Albeit positively charged, colistin is a significantly hydrophobic antibiotic that appears to gain access to the IM by permeating through the OM bilayer, while tetracyclines, chloramphenicol, and quinolones use a lipid-mediated or a porin-mediated pathway depending on protonation state[66]. These antibiotic classes are classified within the same group of molecules according to their interactions with the cell permeability barriers[9]. The fact that other relatively hydrophobic antibiotics such as aminoglycosides are unaffected by the disruption of the Ttg2 system speaks in favor of the observed correlation between membrane-phospholipid content and specific susceptibility to certain antibiotics[52,53]. Another hypothesis that would explain the different impact of Ttg2 disruption on different antibiotic classes would be the possibility that phospholipids carried by the Ttg2 system, particularly cardiolipin, may interact with or stabilize certain efflux pumps in *P. aeruginosa*. A *P. putida* cardiolipin synthesis mutant was more susceptible to several antibiotics and to toluene, probably due to structural alterations in the RND efflux pumps[67]. Indeed, the protein composition of the OM can also have a strong impact on the sensitivity of bacteria to the different antibiotic classes[66]. It was probably the secondary effects of the removal of Ttg2D function that led the authors of another study, based on results on a Ttg2A mutant, to conclude that the function of the Ttg2 system in *P. aeruginosa* was associated with the export of antibiotics such as tetracycline out of the cell[33]. Although further investigations are still required, the activity of the Ttg2 system on membrane-phospholipid homeostasis appears to be partly responsible for the lower basal susceptibility of *P. aeruginosa* to antimicrobial agents, particularly to polymyxins (see Supplementary Discussion).

## Methods

**Bacterial strains**. All bacterial strains used in this study are provided in Supplementary Table 5 and growth conditions in Supplementary Methods.

**Expression, purification, and preparation of Ttg2D$_{Pae}$ protein variants**. Recombinant Ttg2D from *P. aeruginosa* (Ttg2D$_{Pae}$) was obtained in the cytoplasm of *Escherichia coli* BL21(DE3) using the pET-based expression system and was purified to >99% purity. In addition, a *P. aeruginosa* PAO1 mutant lacking *ttg2D* was used for the homologous expression of a His-tagged variant of the protein in its natural environment and its subsequent purification. Periplasmic protein preparation from *P. aeruginosa* was done by the sucrose-lysozyme method[68]. Both protein variants are tagged with a C-terminal 6-histidine tail for purification by

affinity chromatography. Detailed methods for proteins expression and production are available in Supplementary Methods. Samples containing both homologously and heterologously produced proteins were desalted with the appropriate buffer on centricon micro concentrators for MS analyses.

**Ttg2D$_{Pae}$ structure resolution**. Recombinant Ttg2D$_{Pae}$ obtained in *E. coli* was used to produce crystals for structure determination. Methods for crystallization, data collection, and structure refinement are available in Supplementary Methods. The data collection, processing, and refinement statistics are given in Table 1. Atomic coordinates and structure factors have been deposited in the PDB with entry code 6HSY.

**Delipidation of purified recombinant Ttg2D$_{Pae}$**. Recombinant protein produced in *E. coli*, diluted 1:1 with 1% TFA, was delipidated using an HPLC system and a C18 column (Phenomenex Jupiter 5U C18 300A) in 0.1% TFA. Protein was eluted with a gradient of acetonitrile, 0.1% TFA (monitored at 214 and 280 nm) and its delipidation was confirmed by native MS analyses (Fig. 4, inset). Delipidated protein was lyophilized, and typically resuspended in 100 mM NaCl, 10 mM Tris-HCl (pH 8.5) to counteract the acidity of TFA, before exchanging the buffer to the desired one.

**Native MS analysis and identification of bound phospholipids**. Native MS experiments were performed using a Synapt G1-HDMS mass spectrometer (Waters, Manchester, UK) at the Mass Spectrometry Core Facility of IRB Barcelona. Prior to the analysis, samples were desalted with 100 mM ammonium acetate on a centricon microconcentrator. Samples were infused by automated chip-based nanoelectrospray using a Triversa Nanomate system (Advion BioSciences, Ithaca, NY, USA) as interface. After ion isolation, fragmentation was performed by CID (collision induced dissociation) in the transfer or trap region by applying increasing collision energies. See Supplementary Methods for further details. Three technical replicates were performed for native MS experiments.

Analysis of protein Ttg2D$_{Pae}$ variants and their bound phospholipids was also done under denaturing conditions by electrospray ionization MS in both the positive and negative ion mode (see Supplementary Methods). Fragmentation analysis of the most abundant and representative glycerophospholipids released from Ttg2D$_{Pae}$ was done under denaturing MS conditions (non-native) in positive mode. Annotation of most abundant phospholipids present in each sample was done using Lipidomics Gateway (http://www.lipidmaps.org) based on the *m/z* values of MS spectra and according to Oursel et al.[44] and Gidden et al.[45] for *E. coli* phospholipids, and Groenewold et al.[47] for *P. aeruginosa* phospholipids. In addition, to determine the phospholipid composition from both *E. coli* BL21(DE3) and *P. aeruginosa* PAO1, and to identify the most detected phospholipid species by MS, lipidome analysis of intact cells was done according to Angelini et al.[69] (see Supplementary Methods).

**Cardiolipin binding to Ttg2D$_{Pae}$ by native MS**. In all, 1.5 mg of cardiolipin CL (18:0)$_4$ (sodium salt) (Avanti Polar Lipids, Merck, 710334P-25MG) were reconstituted in MeOH, vortexed, and sonicated to obtain a homogeneous solution (1.5 mg/ml, 980 μM). Delipidated protein Ttg2D$_{Pae}$ was prepared for the reaction mixture in 100 mM ammonium acetate at 0.8 mg/mL (35 μM). Delipidated protein was mixed with cardiolipin at 1:9 molar ratio (26.25:245 μM) and incubated at 37 °C for 1 h at 500 r.p.m. Final buffer composition included 75 mM ammonium acetate and 25% MeOH. The excess of cardiolipin were cleaned changing the buffer to 200 mM ammonium acetate using centrifugal filter units of 3K cut-off (Amicon Ultra, Millipore). Buffer was exchanged three times at 15 °C at 13,000 g. The reaction was directly injected for MS analysis. Three technical replicates were measured. To check the behavior of cardiolipin and its fragmentation patterns in the different MS conditions applied, commercial CL(18:0)$_4$ has been used as a standard in all assays.

**Generation of markerless *ttg2* mutants in MDR *P. aeruginosa* strains and complementations**. Markerless *P. aeruginosa* mutants were constructed using a modification of the pGPI-SceI/pDAI-SceI system (Supplementary Fig. 13) originally developed for bacteria of the genus *Burkholderia* and other Gram-negative organisms[49,70]. The bacterial strains and plasmids of the pGPI-SceI/pDAI-SceI system were kindly donated by Uwe Mamat (Leibniz-Center for Medicine and Biosciences, Research Center Borstel, Borstel, Germany) with permission of Miguel A. Valvano (Center for Infection and Immunity, Queen's University, Belfast, UK). The pGPI-SceI-XCm plasmid[48] was first modified to facilitate the generation of *ttg2* mutants in MDR *P. aeruginosa* strains. Plasmid modifications include replacement of the chloramphenicol resistance cassette by an erythromycin resistance cassette and deletion of a DNA region containing the Pc promoter found in *P. aeruginosa* class 1 integrons (Supplementary Methods and Supplementary Table 5 for details). The sequence of the new suicide plasmid vector, pGPI-SceI-XErm, is available through GenBank under the accession number KY368390. For complementation in PAO1, full *ttg2* operon or the codifying region of the *ttg2D* gene were cloned into the broad-host-range cloning vector pBBR1MCS-5 or a variant thereof containing the arabinose promoter, respectively (Supplementary Table 5). For complementation experiments in MDR strains, the cloning vector pBBR1MCS-5 was first modified to confer resistance to erythromycin (see details in Supplementary Methods). Sequence for the new cloning vector, pBBR1MCS-6, is available

through GenBank under the accession number KY368389. Complemented strains were obtained by transforming mutant cells with the corresponding pBBR1MCS derivative plasmid. The expression of *ttg2D* in mutant and complemented strains was verified by RT-PCR and quantitative real-time RT-PCR analysis (Supplementary Methods and Supplementary Fig. 13).

**OM permeabilization assay**. Fluorometric assessment of outer membrane permeabilization was done by the NPN uptake assay as described by Loh et al.[71] with modifications (Supplementary Methods). Since *P. aeruginosa* PAO1 cells have proven to be poorly permeable to NPN[9], either EDTA (0.2 mM) or colistin (10 µg/ml) was added to cells to enhance uptake and fluorescence.

**Susceptibility to antibiotics and membrane-damaging agents**. Antimicrobial susceptibility to a range of antibiotics was tested by determination of the MIC using the broth microdilution method or Etest (Biomerieux) strips, following the Clinical and Laboratory Standards Institute (CLSI) guidelines[72,73] and manufacturer's instructions, respectively (see Supplementary Methods for details). MIC differences higher than twofold were considered significant changes in antibiotic susceptibility. Low-level, basal resistance to a given antibiotic was defined as that of an organism lacking acquired mechanisms of resistance to that antibiotic and displaying a MIC above the common range for the susceptible population[74]. Clinical susceptibility breakpoints against *Pseudomonas* sp. for selected antibiotics have been established by EUCAST[75]. Tolerance to organic solvents and SDS/EDTA was assessed using solvent overlaid-solid medium and MIC assays, respectively (Supplementary Methods).

**Biofilm formation**. Biofilm quantification in a 96-well microtiter plate by the crystal violet assay was done as previously described[76] with modifications (Supplementary Methods).

**Bioinformatic analysis**. Details are provided in the Supplementary Methods.

**Statistics and reproducibility**. For MS assays, experiments were carried out in triplicate. Phenotypic assays were confirmed by at least three independent replicates and the mean with the standard deviation is shown in figures. A one-way ANOVA with Tukey's multiple comparison test was used to determine the significance of the data between groups.

**Reporting summary**. Further information on research design is available in the Nature Research Reporting Summary linked to this article.

## Data availability

The X-ray crystal structure and structure factors of Ttg2D$_{Pae}$ have been deposited in Protein Data Bank under accession codes PDB 6HSY. Nucleotide sequences for plasmids pGPI-SceI-XErm and pBBR1MCS-Erm are available on the GenBank database with the accession numbers KY368390 and KY368389, respectively. Source data underlying graphs and charts are provided for download as Supplementary Data 1. All other data that support the findings of this study are available from the corresponding author upon reasonable request.

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

## Acknowledgements

This work has been supported by funding under the Seventh Research Framework Programme of the European Union (ref. HEALTH-F3-2009-223101) and the Spanish Ministry of Science, Innovation and Universities (ref. BIO2015–66674-R). The funders had no role in study design, data collection and interpretation, or the decision to submit the work for publication. We acknowledge the European Synchrotron Radiation Facility for provision of synchrotron radiation facilities and thank the staff of ID23-1 for assistance in using the beamline. Part of the mass spectrometry experiments were performed at UAB's proteomics facility SePBioEs.

## Author contributions

D.Y., M.D.-L., L.C., O.C.-S., A.M., M.F.-N. and M.V. conducted the experiments; D.Y., M.D.-L., L.C., O.C.-S., M.V., I.G. and X.D. designed the experiments and participated in the analysis and interpretation of experimental data; D.Y., L.C., O.C.-S. and M.D.-L. wrote the paper; M.V., I.G. and X.D. supervised research and revised the manuscript.

## Competing interests

The authors declare no competing interests.
