## [Peer Review File · Communications Biology]

Reviewers' comments:

Reviewer #3 (Remarks to the Author):

The authors have revised the manuscript to a sufficiently high quality and have address the reviewers' concerns. In my estimation, the manuscript is now suitable for publication.

Reviewer #4 (Remarks to the Author):

This manuscript explains the structural and functional characterisation of the *Pseudomonas aeruginosa* Ttg2D protein. My comments are focused on native mass spectrometry, ABC proteins and protein-lipid interactions with relevance to my expertise.

I find title of the article very misleading, Ttg2 is a SBP not ab ABC system.

The manuscript is overall well-written and authors have used a variety of techniques to demonstrate their claim. It is very clearly demonstrated that the protein has endogenous lipids co-purified with the protein. After delipidation, cardiolipins (CLs) are shown to make a complex with the protein. Addition of CLs is however irrelevant considering it is a bacterial protein. It will be helpful for readers to understand the conditions used for protein-CLs complex formation. I am not happy with the methods used for protein delipidation, there is other 'gentle' ways to remove the lipids instead of treating with FA.

Considering it's a SBP protein, as the name suggests the function has to deal with transfer of substrate to the ABC transporter, as known for other SBPs. Can authors provide evidence of any other SBP that has a role in lipid transfer? the refence Mla pathway is irrelevant.

Soluble proteins do interact with lipids and it's shown previously by research articles from the group of Carol Robinson. In my experience it is very unlikely for a protein of 22 kDa to transfer a lipid molecule.

Referring to Fig 3 (a), for soluble proteins highly charged state mass spectrum is not very uncommon but having both high charge state and low charge state envelopes of the same protein is a very rare observation. Perhaps it indicates that the population of the protein is not properly folded/structured. I'd suggest a CD spectroscopy experiment would be helpful to establish proper folding state of the protein.

Fig 3 (b). it is little confusing, first for clarity, authors could have shown 2701.18 m/z peak after selection (without any activation). Once 2701.18 (9+) charge state is activated, it is unclear if the lipid is stripped away as a neutral loss or it takes a charge away from the protein. If there is no neutral loss, then it's impossible to see an apo protein peak for 9+ charge state. For apo protein charge state 8+ makes sense as it arises after activation of 9+ (protein-lipid) complex. Would be helpful to get authors comments on this point. Also, there is no need to label the peak with (M+9H), labelling z=9 or 9+ is good enough.

I'd suggest that the article should describe that the Ttg2 is SBP is protein involved in interactions with lipids. To propose it acts as a lipids transporter is a very big claim which is not proven in the article.

Changes made as a result of comments from Reviewer 4 are highlighted in yellow in both the manuscript and supplementary file.

Reviewer #4 (Remarks to the Author):

This manuscript explains the structural and functional characterization of the *Pseudomonas aeruginosa* Ttg2D protein. My comments are focused on native mass spectrometry, ABC proteins and protein-lipid interactions with relevance to my expertise.

I find title of the article very misleading, Ttg2 is a SBP not an ABC system.

Ttg2 is actually an ABC system (Ttg2D is its periplasmic soluble SBP component). However, we understand the objection of the reviewer to the title, since we present results for Ttg2D, not for the entire system. We have accordingly changed the title to "The *Pseudomonas aeruginosa* substrate-binding protein Ttg2D functions as a general glycerophospholipid transporter across the periplasm".

The manuscript is overall well-written and authors have used a variety of techniques to demonstrate their claim. It is very clearly demonstrated that the protein has endogenous lipids co-purified with the protein. After delipidation, cardiolipins (CLs) are shown to make a complex with the protein. Addition of CLs is however irrelevant considering it is a bacterial protein.

We do not quite understand why "addition of CLs is however irrelevant considering it is a bacterial protein". The reviewer seems to suggest that bacteria do not have cardiolipins.

In fact, in most Gram-positive and Gram-negative bacteria the membranes are composed of different phospholipid classes including cardiolipin species, with a variable proportion among bacterial species. Thus, *P. aeruginosa* has ca. 11% cardiolipin in its membranes. As mentioned in the Discussion section, in Gram-negative bacteria, cardiolipin is mostly located within the inner membrane (the site of its synthesis), but it is also present in the outer membrane where it facilitates proper localization of proteins on the bacterial surface. The finding that the Ttg2 system (of which Ttg2D is a component) can transport cardiolipin through the periplasm is relevant since until now a mechanism of retrograde transport of cardiolipin had not been described. Cardiolipin forms membrane domains in bacterial cells that participate in diverse cellular functions. It is well-known that an imbalance in the cardiolipin content of the outer membrane could cause an increase in susceptibility to cationic antibiotics (e.g. Raetz CR, et al. *J Bacteriol* 1979, 139: 544-551).

It will be helpful for readers to understand the conditions used for protein-CLs complex formation. I am not happy with the methods used for protein delipidation, there is other 'gentle' ways to remove the lipids instead of treating with FA.

Concerning the conditions used for protein-CLs complex formation, we have now improved the description of the protocol in the Methods section. After delipidation the protein buffer was exchanged to 100 mM ammonium acetate. Sodium cardiolipin was dissolved in methanol and mixed with the protein solution at a 9:1 molar ratio. Under these conditions, lipid-free Ttg2D_{Pae} was able to bind free cardiolipin (Figure 4), as demonstrated by native MS.

With regard to the method used for protein delipidation, other authors have also used harsher protocols to remove PL from the MlaC protein (the Ttg2D ortholog in *E. coli*). For instance, Ercan *et al.* (Biochemistry 2018, referenced in the manuscript) generated PL-free forms of MlaC for PL transfer experiments. They purified the protein under denaturing conditions (8 M urea), and then refolded it by buffer-exchange steps. After these, they demonstrated that PL could be spontaneously transferred from MlaC to members of the Mla pathway, and that MlaC had a higher binding affinity for PL.

We know that our delipidation protocol produces a stable, folded Ttg2D because we were able to co-crystallize it with cardiolipin (see figure below). The crystallization of the protein, which happened in the same space group and with almost identical cell parameters as the Ttg2D structure described in this study, is considered an unequivocal sign of stability (e.g. Deller *et al. Acta Crystallogr F Struct Biol Commun* 2016, 72: 72-95). Note that the crystals were obtained quite before the MS experiments were performed, and we used the cardiolipin species that we had at hand at that moment (bovine-heart cardiolipin). Since the crystals obtained did not diffract to a sufficient resolution to solve the structure of the complex and the yield of delipidation was on the low side for crystallography experiments, we turned to the MS experiments to demonstrate binding, using a cardiolipin, CL(18:0)₄, closer to the species found in *P. aeruginosa*. We did not include this crystallization data in the manuscript because it is a dead end and, in our opinion, the MS results, demonstrating binding, should be sufficient to show that the delipidated protein was in its active form. We could add this Figure in supplementary material if the editor finds it necessary (we have not done so in the current revised version).

Figure: Delipidated Ttg2D co-crystallized with bovine-heart cardiolipin. Left panel: *In situ* diffracting crystal. Right panel: One diffraction image used to determine the data collection strategy (1s exposure, 1° oscillation). Crystals of delipidated Ttg2D with cardiolipin diffracted at ~3.8 Å resolution with the same space group (P3221) and almost identical cell parameters as the deposited Ttg2D structure (PDB 6SHY).

Considering it's a SBP protein, as the name suggests the function has to deal with transfer of substrate to the ABC transporter, as known for other SBPs. Can authors provide evidence of any other SBP that has a role in lipid transfer? the refence Mla pathway is irrelevant.

The reference to the Mla pathway is not irrelevant. As extensively described in the Introduction, Ttg2 is the ortholog in *P. aeruginosa* of the Mla ABC transport system described in *E. coli* and other Gram-negative bacteria. The naming difference is of purely historic nature and, as a matter of fact, we could have perfectly used the Mla nomenclature also for *P. aeruginosa* (as a curiosity, the Ttg name comes from a work on *P. putida* that related this and 5 other operons, hence the numbering 2, with resistance to toluene; later, it was seen that this operon was the ortholog of Mla in *E. coli*). The components of this system in *P. aeruginosa* are Ttg2A, Ttg2B, Ttg2C, Ttg2D, Ttg2E and VacJ. Ttg2D is the soluble periplasmic substrate-binding protein component of this system and is the ortholog of MlaC in *E. coli*. To clarify the evolutionary relationship between the Mla/Ttg2 ortholog systems and their components in different bacteria, we have added the following new panel to Figure 2 (amino acid identities (%) between ortholog pairs as determined by Clustal Omega are shown on a gray background).

The role of MlaC in phospholipid transfer to other components of the ABC transport system has been extensively demonstrated (Hughes GW, *et al.* 2019. *Nat Microbiol* 4, 1692-1705. and Ekiert DC, *et al.* 2017. *Cell* 169, 273-285 e217). We should not need to stress that the Ttg2D structure we are presenting in this manuscript shows density corresponding to two phospholipids (with higher probability than one cardiolipin) in the binding site. We already explained in our response to the reviewers from *Nat Commun* that, despite the electron density for the phospholipids is not very good, the unbiased 2mFo - DFc electron density map (Fig. S2 A) clearly indicates that Ttg2D binds four acyl chains in the crystal. The quality of the FEM map contoured at 1.5 σ (Fig. 1A and Fig. S2 B) and real-space correlation coefficients of 0.9 for the lipids indicate that they are correctly modeled. Binding of phospholipids to a binding site that is specific for a certain type of molecular species (as SBPs are) cannot be interpreted in terms of co-purification, as suggested later by the reviewer. As we also mention in the manuscript, this is not the only structure of an MlaC ortholog that has been determined with phospholipids in the binding site, which are captured by the protein from the production medium. As a matter of fact, it is relatively common for SBPs to be crystallized with their substrate, if this is present in the production medium.

In summary, a substrate-binding protein that binds phospholipids specifically (i.e. in its binding site), that captures them spontaneously from the medium in a fixed stoichiometry (one cardiolipin as found by MS, four acyl tails, more likely from two diacyl phospholipids, as found in the crystal) and belongs to a transport system whose ortholog in *E. coli* has been shown to transport phospholipids, can be safely assumed to be a phospholipid transporter. This is not even the point of this manuscript, since it is generally accepted by the community that MlaC is a phospholipid transporter in the different species (including *P. aeruginosa*) in which the Mla operon shown above exists. The main points of our study, clarifying important

open questions for this transporter, are that 1) in *P. aeruginosa* Ttg2D can transport two diacyl phospholipids together, a fact that had been questioned by the community (see for example the comments of the reviewers to the Nat Commun submission), who thought that only one molecule would bind at a time and when four acyl chains were present in the cavity this had to be, therefore, a cardiolipin, and 2) that the variety of glycerophospholipids that can be transported is large, including a variety of diacyl and tetraacyl ones (hence the title, *The Pseudomonas aeruginosa* substrate-binding protein Ttg2D functions as a general glycerophospholipid transporter across the periplasm).

Soluble proteins do interact with lipids and it's shown previously by research articles from the group of Carol Robinson. In my experience it is very unlikely for a protein of 22 kDa to transfer a lipid molecule.

There are several examples of small soluble human proteins that carry phospholipids and other lipid molecules. For example, SCP-2 (PDB 1C44) is a 13.3 kDa protein with the ability to transfer sterols and a variety of other lipids between membranes. The human glycolipid transfer protein GLTP (25 kDa; PDB 4H2Z) catalyzes the transfer of various glycosphingolipids between membranes. There are other examples in both prokaryotes and eukaryotes that are listed in Wong LH, Čopič A and Levine TP. Trends Biochem Sci 2017, 42: 516-530.).

Molecular weight aside (for both transporter and substrate), the Ttg2/Mla pathway contains similar elements to those of the ABC transport system Lol (localization of lipoproteins) in *E. coli*. LolA is the soluble SBP that has been shown to transport mature lipoproteins across the periplasm (Yakushi, T. et al. Nat Cell Biol 2000, 2: 212-218). LolA has a hydrophobic cavity that represents a possible binding site for the lipid moiety of the lipoproteins (Takeda K. et al. EMBO J 2003, 22: 3199-3209).

Referring to Fig 3 (a), for soluble proteins highly charged state mass spectrum is not very uncommon but having both high charge state and low charge state envelopes of the same protein is a very rare observation. Perhaps it indicates that the population of the protein is not properly folded/structured. I'd suggest a CD spectroscopy experiment would be helpful to establish proper folding state of the protein.

The reviewer is right. We have a population of partially denatured species in the solution. However, we are convinced that it is due to the previous buffer exchange of the sample required to do Native MS to get rid of interfering non-volatile salts and to use compatible buffers for the direct MS measure. Moreover, this partially denatured population is independent of the folded protein that still remains in solution and from which we have derived conclusions. Additionally, and despite using very soft ionization conditions and modified backpressures to preserve the non-covalent complexes, there are still protein complexes with low affinity that inevitably tend to further partially dissociate and in some spectra the unbound folded protein can also be detected as a minor species.

With regard to a proper folding state of protein Ttg2D prior to sample preparation, we should stress that this is the same protein species (production in *E. coli*) we crystallized and solved the structure for (PDB 6SHY).

Fig 3 (b). it is little confusing, first for clarity, authors could have shown 2701.18 m/z peak after selection (without any activation). Once 2701.18 (9+) charge state is activated, it is unclear if the lipid is stripped away as a neutral loss or it takes a charge away from the protein. If there is no neutral loss, then it's impossible to see an apo protein peak for 9+ charge state. For apo protein charge state 8+ makes sense as it arises after activation of 9+ (protein-lipid) complex. Would be helpful to get authors comments on this point.

To study the complexes between the protein and 2 phospholipids or 1 cardiolipin molecule, the ion m/z 2430 and m/z 2701, which corresponds to charge 10 and 9 of the complex, respectively, were selected and isolated. In both studies, the transfer collision energy was gradually increased from 4V to 50-60V. Figure S8 of Supplementary Materials shows the experiment of isolation of the m/z 2701 peak (9+ complex) and the increasing of the collision energy (Figure S8A). We have prepared a new figure that shows the species detected when the isolated m/z 2430 peak (10+ complex) was gradually disrupted increasing the transfer CE (Figure S8B).

When the transfer CE is 4V, we were not applying enough energy to release the molecules that were bound to the protein. However, with the transfer CE of 35V, we observed that the complex between the protein and the different ligand species started to disrupt and lipids were released and detected as charge 1 in the mass spectrum zone of m/z 500-800, as it is shown in Figure 3C. The more collision energy we applied to the protein-lipid complex, the more disruption of that complex and release of binding molecules we detected. With a Transfer CE of 50-60V, we only identified the free protein.

In both experiments, we observed that the ligand could be released as neutral loss and as charge +1. Yin et al (Yin, Xie and Loo, J Am Soc Mass Spectrom. 2008) published separate CAD experiments for RNase A (13682 Da) complexed to 2'-CMP (323 Da), CDP (403 Da), and CTP (483 Da) in pH 6.6 solution. The charge state of the most abundant protein-ligand complex molecule was 8+ for each of the ligands studied. The authors showed the MS/MS spectra of the 8+ RNase A complex with 2'-CMP, CDP, and CTP. Dissociation of the $(M + 8H + \text{CMP})^{8+}$ complex (where "M" is the RNase A protein) yields the complementary $(M + 7H)^{7+}$ and $(\text{CMP} + H)^+$ product ion pair. In addition, the $(M + 8H)^{8+}$ apo-protein with concurrent loss of neutral 2'-CMP was observed as well. Furthermore, dissociation of the $(M + 8H + \text{CDP})^{8+}$ complex yields the complementary $(M + 7H)^{7+}$ (m/z 1955) and $(\text{CDP} + H)^+$ (m/z 404) product ion pair as one dissociation pathway, and a minor pathway yielding the $(M + 8H)^{8+}$ protein with loss of neutral CDP. Their results support what we observe in our system. In our case, the lipid is released as a neutral and as a +1 charged compound. More specifically, the dissociation of the $(P+2PL)^{9+}$ complex (m/z 2700 peak) yields the complementary $(P+1PL)^{8+}$ and $(PL)^+$ product ion pair, but also the $(P+1PL)^{9+}$ with concurrent loss of neutral PL. In addition, the $(P+1PL)^{9+}$ complex dissociates into $(P)^{8+}$ and $(PL)^+$ ions. Furthermore, the $(P)^{9+}$ apo-protein with concurrent loss of neutral PL was detected as well (Figure S8A). MS/MS of the $(P+2PL)^{10+}$ complex (m/z 2431 peak) shows identical behavior (Figure S8B).

Figure S8. Complexes formed by protein Tgt2D_{Pae} produced in *E. coli* (P) and phospholipids (PL). The dissociation of the (P+2PL)⁹⁺ complex (m/z 2700 peak) yields the complementary (P+1PL)⁸⁺ and (PL)⁺ product ion pair, but also the (P+1PL)⁹⁺ with concurrent loss of neutral PL. In addition, the (P+1PL)⁹⁺ complex dissociates into (P)⁸⁺ and (PL)⁺ ions. Furthermore, the (P)⁹⁺ apo-protein with concurrent loss of neutral PL was detected as well. MS/MS of the (P+2PL)¹⁰⁺ complex (m/z 2430 peak) shows identical behavior. (A) Zoom of the 2300–3100 m/z range of the fragment ion mass spectra of isolated ion m/z=2700 (z=9) at varying transfer collision energies (CE). Isolation and dissociation of the species P+2PL at CE greater than 30V have allowed the detection of species with one ligand (P+1PL). However, at CE of 50V there still exists a population of fully charged species were the 2PL could correspond to CL-like molecules. Increasing the CE results in complete release of the phospholipids. The mass of the bound molecules can then be calculated from the mass difference between the bound and unbound peaks. (B) Zoom of the 2000–3000 m/z mass spectra range of fragmentations of isolated ion m/z=2430 (z=10) at varying transfer collision energies (CE). Increasing the CE

results in partial release of the bound binding ligands. Note that 2PL could be a mixture of populations with two different phospholipids or one cardiolipin molecule.

Moreover, we also observed similar behavior in the native MS/MS experiment of the protein-cardiolipin complex formed between delipidated Ttg2D_{Pae} and commercial cardiolipin CL(18:0)₄. Dissociation of the (P+CL)¹⁰⁺ complex yields the complementary (P)⁹⁺ and (CL)⁺ product ion pair as one dissociation pathway, and the (P)¹⁰⁺ protein with loss of neutral CL as another pathway (Figure S11).

On the other hand, we observed that the nature of the lipids that could bind the protein is very diverse. They could be Phosphoethanolamine (PE), Phosphatidylglycerol (PG), Phosphatidylserine (PS) and Phosphatidylglycerol-phosphate (PGP) (Table S4 of Supplementary Materials) and also cardiolipins. Therefore, protein-lipid complexes are formed between protein and different families of lipids. Li et al (Li, Heitz, Le Grimellec and Cole, Rapid Communications in Mass Spectrometry, 2003) reported the stability *in vacuo* of lipid-peptide complexes. Authors worked with peptide-lipid complexes formed between peptide P294 and phosphatidylglycerols (PGs; dilauryl = DLPG and dimyristyl = DMPG) or phosphatidylcholines (PCs; DLPC and DMPC). They proved that peptide-lipid and protein-lipid complexes detected by ES-MS already existed in solution. For that reason, we are sure that the protein-lipid complexes detected in our study in gas phase were also present in solution.

Also, there is no need to label the peak with (M+9H), labelling z=9 or 9+ is good enough.

We relabeled the peaks of panels 3A and 3B (Fig. 3) as suggested.

I'd suggest that the article should describe that the Ttg2 is SBP is protein involved in interactions with lipids. To propose it acts as a lipids transporter is a very big claim which is not proven in the article.

This has been discussed above.

REVIEWERS' COMMENTS:

Reviewer #4 (Remarks to the Author):

Thank you for considering my suggestion, I believe article title now presents a true reflection of the target protein.